# Modal Parameter Recursive Estimation of Concrete Arch Dams under Seismic Loading Using an Adaptive Recursive Subspace Method

**DOI:** 10.3390/s24123845

**Published:** 2024-06-14

**Authors:** Xinyi Zhu, Jianchun Qiu, Yanxin Xu, Xingqiao Chen, Pengcheng Xu, Xin Wu, Shaolong Guo, Jicheng Zhao, Jiale Lin

**Affiliations:** 1College of Hydraulic Science and Engineering, Yangzhou University, Yangzhou 225009, China; 213210833@stu.yzu.edu.cn (X.Z.); m18994113495@163.com (P.X.);; 2Modern Rural Water Resources Research Institute, College of Hydraulic Science and Engineering, Yangzhou University, Yangzhou 225009, China; 3State Key Laboratory of Hydrology-Water Resources and Hydraulic Engineering, Hohai University, Nanjing 210098, China; xu_yanxin@hhu.edu.cn (Y.X.); 200402020001@hhu.edu.cn (X.C.);; 4National Engineering Research Center of Water Resources Efficient Utilization and Engineering Safety, Hohai University, Nanjing 210098, China; dream2020@tcu.edu.cn; 5College of Water-Conservancy and Hydropower, Hohai University, Nanjing 210098, China; 6College of Civil Engineering, Tianjin Chengjian University, Tianjin 300384, China

**Keywords:** concrete arch dam, modal parameter recursive estimation, damage identification, adaptive recursive subspace method

## Abstract

Modal parameter estimation is crucial in vibration-based damage detection and deserves increased attention and investigation. Concrete arch dams are prone to damage during severe seismic events, leading to alterations in their structural dynamic characteristics and modal parameters, which exhibit specific time-varying properties. This highlights the significance of investigating the evolution of their modal parameters and ensuring their accurate identification. To effectively accomplish the recursive estimation of modal parameters for arch dams, an adaptive recursive subspace (ARS) method with variable forgetting factors was proposed in this study. In the ARS method, the variable forgetting factors were adaptively updated by assessing the change rate of the spatial Euclidean distance of adjacent modal frequency identification values. A numerical simulation of a concrete arch dam under seismic loading was conducted by using ABAQUS software, in which a concrete damaged plasticity (CDP) model was used to simulate the dam body’s constitutive relation, allowing for the assessment of damage development under seismic loading. Utilizing the dynamic responses obtained from the numerical simulation, the ARS method was implemented for the modal parameter recursive estimation of the arch dam. The identification results revealed a decreasing trend in the frequencies of the four initial modes of the arch dam: from an undamaged state characterized by frequencies of 0.910, 1.166, 1.871, and 2.161 Hz to values of 0.895, 1.134, 1.842, and 2.134 Hz, respectively. Concurrently, increases in the damping ratios of these modes were observed, transitioning from 4.44%, 4.28%, 5.42%, and 5.56% to 4.98%, 4.91%, 6.61%, and 6.85%%, respectively. The correlation of the identification results with damage progression validated the effectiveness of the ARS method. This study’s outcomes have substantial theoretical and practical importance, facilitating the immediate comprehension of the dynamic characteristics and operational states of concrete arch dam structures.

## 1. Introduction

As an important dam type, concrete arch dams, which are constructed worldwide, have notable advantages, including robust structural stability, savings in construction materials, heightened safety, and formidable flood resistance [1,2,3]. As one essential component of society’s infrastructure system, concrete arch dams play the roles of flood control, power generation, irrigation, shipping, and water supply. Many of these existing arch dams are located in seismically active regions, and their seismic safety has been one of the crucial issues in the design and construction stages for researchers because of potential damage or failure that threatens public safety and economic infrastructure [4,5]. Several instances of dam damage sustained due to strong earthquakes have been recorded, including in the Hsinfengkiang concrete dam [6] in China, the Koyna gravity dam [7] in India, and the Pacoima arch dam [8] in the USA. To ensure the safe operation of arch dams under seismic loads, timely damage identification is crucial [9,10,11].

Conventional methods for post earthquake damage identification in concrete dams involve analyzing static monitoring data such as deformation, seepage observations, and inspection reports. However, these methods may not be sensitive to certain types of damage, such as localized cracks in concrete dam structures, especially in underwater and internal components. Some nondestructive testing techniques, such as ultrasonic inspection [12], ground-penetrating radar detection [13], and acoustic emission detection [14], generally require prior knowledge of the damaged areas for structural damage identification. However, these nondestructive methods are more effective for small concrete structures and face challenges in identifying damage in large and complex concrete arch dam structures. Utilizing the measured dynamic response of concrete arch dams under seismic loads to identify changes in structural damage states can address the aforementioned shortcomings, thereby enhancing the capability and real-time nature of damage identification for these dams. Structural damage identification of concrete arch dams based on dynamic characteristics has the advantages of being nondestructive, multilevel, and multiscale. To achieve the goal of identifying arch dam damage using dynamic characteristics, seismic monitoring should be conducted for arch dams. In recent years, seismic monitoring for arch dams has been proposed by some design codes. For example, the code entitled ‘Technical Specification of Strong Motion Monitoring for Seismic Safety of Hydraulic Structures’ (guideline number SL 486-2011) [15] in China provided clear provisions for the arrangement and operational management of strong seismic monitoring instruments for arch dams in high-intensity seismic areas. Meanwhile, the code entitled ‘Technical Specifications for Safety Monitoring of Concrete Dams (guideline number SL601-2013)’ [16] in China mandated that dynamic response monitoring instruments must be installed in Grade 1 dams designed for a seismic intensity of VII or higher and in Grade 2 dams designed for a seismic intensity of VIII or higher. Following the requirements of these technical specifications, there has been a growing trend among concrete arch dams in high-seismic-intensity areas to implement the setup of seismic observation instruments. Based on the measured dynamic responses of concrete dams, the damage to arch dams can be identified by analyzing variations in structural dynamic properties [17]. Therefore, conducting seismic monitoring and damage identification of arch dams is highly important. This contributes to a better understanding of the working conditions of concrete dams, ensures the safety of arch dams, and promotes both theoretical understanding and practical engineering applications.

Hitherto, various vibration-based methods have been developed for post earthquake structural damage identification. Many of these methods focus on the identification of modal parameters, as modal parameters, including the natural frequency, mode shape, and damping ratio, play a crucial role in revealing the dynamic characteristics of a structure [18,19]. These parameters are fundamental indicators that provide insights into how a structure responds and behaves under dynamic loading conditions. They provide inherent features of the structure, carry clear physical significance, and serve as vital reference points for structural damage identification, model refinement, and the optimization design of structural dynamic characteristics [20,21,22]. 

Modal parameters are influenced by factors such as the structural form, mass distribution, structural stiffness, material properties, and structural connections. When subjected to seismic loads, an arch dam may experience localized structural damage, such as cracking. This imperils the overall integrity of the structure, resulting in a decrease in structural stiffness and ultimately causing alterations in the dynamic characteristics of the dam [23,24]. Due to the inevitable decrease in stiffness and increase in damping during the occurrence and development of structural damage, changes in structural modal parameters are inevitably induced. Based on the measured dynamic responses, the identification of modal parameters allows for the comparison of parameter changes at different times, facilitating the determination of whether there has been a change in the structural state [25,26,27]. 

During the normal operation of a concrete arch dam, the inherent modal parameters of the structure do not change with time and can be identified using the aforementioned time-invariant structural modal parameter identification methods. However, in another scenario, as an arch dam experiences damage from seismic loading, and with the gradual accumulation and progression of damage, the modal parameters of the arch dam change over time [11,28]. In such cases, a time-varying modal parameter (TMP) identification method needs to be employed. 

Currently, there are two main methods for identifying TMPs. One type is the segmented analysis method [29], which divides response data into small time segments, treating the modal parameters as time-invariant within each segment. The modal parameters identified in each segment are then fitted with a curve to obtain the time-varying patterns. However, this method does not utilize data from previous time segments when estimating parameters for subsequent segments, requiring the selection of shorter time segments for systems with rapidly changing parameters, leading to computational challenges. Another commonly used method is the recursive method [30], which continuously integrates new data while gradually forgetting old data. In the recursive method, the estimated values of the modal parameters are continually updated at each instant. The recursive method offers advantages such as low storage requirements and quick computations, making it beneficial for the rapid diagnosis of concrete dam structures under seismic loads. The recursive methods used for TMP identification include time-varying time series model-based recursive methods and recursive subspace (RS) methods. Recursive tracking techniques such as recursive least squares and recursive maximum likelihood are often used in time-varying time series model-based recursive methods, and they have achieved successful application. For example, Liu L et al. [31] proposed a time-varying multivariate autoregressive model and utilized recursive least squares for estimating the time-varying frequencies and damping ratios. The validation process involved testing the model on a two-degree-of-freedom time-varying vibrational system. Zhi-Sha Ma et al. [32] proposed a kernel ridge regression functional series time-dependent autoregressive moving average (FS-TARMA) recursive identification method. Subsequently, they applied this method to identify the time-varying modal parameters of a numerical three-degree-of-freedom time-varying structural system and a laboratory time-varying structure, which featured a simply supported beam with a moving mass sliding on it. Through Monte Carlo experiments, the method was compared against an existing recursive pseudo linear regression FS-TARMA approach, demonstrating its ability to accurately track the time-varying dynamics in a recursive manner. 

Another commonly used method for identifying the TMP or time-varying parameters is the RS method [33,34], which was developed from the subspace identification method. The subspace identification method used for modal parameter identification requires minimal prior knowledge of the structural system and can be applied to describe complex multi-input–multi-output dynamic systems. If the segmented subspace method is utilized for TMP identification, each small time segment entails a substantial system Hankel dimension, potentially influencing the identification speed during singular value decomposition. However, concrete arch dams are often large and complex and involve high-order structural dynamic models [35]. For the TMP identification of arch dams, considering real-time parameter tracking and recursiveness, a faster and less computationally demanding recursive identification method must be sought. In this scenario, the RS method, due to its fast computation speed and minimal data requirements, presents a significant advantage over other methods in the identification of time-varying parameters in structural systems. Some researchers have employed RS techniques to achieve TMP identification or time-varying parameter identification. For example, Ni ZY et al. [36] proposed a recursive predictor-based subspace method to identify the TMP of a spacecraft with flexible appendages. Compared to the traditional projection approximation subspace tracking recursive algorithm, this method has been demonstrated to possess superior computational efficiency and robustness. Kameyama K et al. [37] proposed a new subspace method for predicting time-varying stochastic systems. The core of this method involved introducing the concept of angles between past and present signal subspaces and interpreting changes in the extended observability matrix as a rotation of the signal subspace. To predict the future signal subspace, the angular velocity and acceleration of the signal subspace were derived. Wen J et al. [38] proposed an RS method with a forgetting factor for on-line system parameter estimation and damage detection using both input and output response measurements. This method can clearly identify abrupt changes in system characteristics, as verified by tests on a three-story steel frame with abrupt changes in interstory stiffness on a shaking table. Additionally, they developed a recursive stochastic subspace identification method for continuously monitoring systems using only stochastic output responses, which was validated using a two-story reinforced concrete frame subjected to low-level white-noise excitation. Huang S et al. [39] proposed an RS method with a fixed window for the time-varying dynamic characteristic identification of seismically excited structures, and the effectiveness of the method was verified by a numerical simulation example of an eight-story shear-type frame and an experimental example of a ten-story RC frame. Ni ZY et al. [40] proposed an improved robust RS method to identify the time-varying modal frequencies of solar panel arrays with different rotation speeds, and the identification results indicated that the proposed method achieved superior time-varying tracking performance compared to traditional recursive methods. Wei Q et al. [41] proposed a method addressing the shortcomings of stochastic subspace identification (SSI) in identifying structural time-varying modal parameters, such as poor computational efficiency and susceptibility to spurious modes. Covariance-driven SSI was enhanced with randomized singular value decomposition and subspace iteration to reduce the computational cost and avoid accuracy loss, forming the lightweight SSI (lwSSI). The effectiveness of the proposed lwSSI-based method was verified by a simulation example of a gravity dam and a steel cantilever beam with time-varying modal parameters caused by mass distribution. Considering the time-varying characteristics exhibited by concrete arch dams under strong seismic loading during the emergence and development of structural damage, and recognizing the significant contributions of modal parameters in reflecting the inherent features of the structure and providing a crucial reference for structural damage identification, model refinement, and optimization of structural dynamic characteristics, the application of the RS method for identifying the TMP in large concrete arch dams has become a crucial research topic. 

In the recursive identification process of the TMP using the RS method, a duration of input–output response data is necessary. As the recursive identification progresses, new input–output data obtained at each moment are consistently integrated, while old input–output data are systematically removed and disregarded. Consequently, the structural modal parameters are continuously refined at each moment. It is evident that data from the current moment and those closer to it are more reflective of the current state compared to data from further back. Therefore, in the RS method, a forgetting factor is often introduced to express the varying impact of data from different time points on the current moment, distinguishing between newer and older data. However, a fixed forgetting factor is commonly employed for slow-variance systems [35]. Concerning the selection of the forgetting factor, a larger value can expedite convergence in slow-variance systems; nevertheless, in the presence of a substantial change in the system subspace, the tracking performance of these methods may experience a notable decline. In contrast, a relatively smaller forgetting factor can enhance the tracking capability of fast-varying parameters.

To address the low efficiency of the conventional RS method in tracking the modal parameters of concrete arch dams, an ARS method based on variable forgetting factors was proposed in this study. In the ARS method, the variable forgetting factors were adaptively updated by assessing the change rate of the spatial Euclidean distance of adjacent modal frequency identification values. Through the introduction of the variable forgetting factor, an unbiased updating form for the Hankel matrix was derived. Hankel matrices with variable forgetting factors, as well as generalized observable matrices and system matrices with variable forgetting factors, were established and derived to achieve the adaptive recursion of the subspace method with variable forgetting factors. The recursive updating of the generalized observability matrix was realized through the orthogonal subspace tracking algorithm, thereby enabling the recursive identification of the system’s state–space matrix. From the eigenvalue decomposition of the system’s state–space matrix, the identification of time-varying modal parameters was achieved. To validate the effectiveness of the proposed ARS method in conducting the recursive identification of modal parameters of arch dams, a concrete arch dam with a maximum height of 305.0 m was taken as a numerical simulation case, utilizing the finite element analysis software ABAQUS for the analysis. In the numerical simulation, the dam body employed Rayleigh damping, and the dynamic water pressure was simulated using the added mass method. The constitutive relationship of the dam body was modeled using a CDP model. Consequently, the occurrence and development of damage in the dam body can be intuitively observed through the tensile damage factor and compressive damage factor across the entirety of the dam body. The proposed ARS method was then used to conduct the modal parameter recursive estimation of the arch dam, and the identification results were consistent with the evolution of damage, thus verifying the effectiveness of the proposed method. Therefore, the proposed ARS method can achieve modal parameter recursive estimation for arch dams and can be used to identify structural damage and its development. The research findings of this study have significant theoretical and practical engineering value.

This paper addresses the limitations of recursive subspace identification methods in modal parameter recursive estimation. By utilizing the spatial Euclidean distance between identified modal frequencies at adjacent time points, a variable forgetting factor (VFF) calculation method was proposed. A Hankel matrix with a forgetting factor was constructed, and expressions for the VFF generalized observability matrix and system matrix were derived. An ARS method with VFF for the modal parameters of arch dam structures was then introduced, enabling the recursive identification and analysis of the modal parameters of concrete dam structures based on dynamic response. The remainder of this paper is organized as follows: Section 2 describes the ARS method with VFF. Section 3 describes how a dynamic simulation of a concrete arch dam was conducted, through which the damage pattern and damage development of the arch dam were obtained, and how a modal parameters recursive estimation was conducted for the arch dam. Finally, Section 4 presents the conclusions drawn from this study.

## 2. ARS Method Based on Variable Forgetting Factors

The RS identification method was developed based on the conventional subspace identification method [33,42]. This approach utilized an orthogonal subspace tracking algorithm to track the observability matrix of the system, avoiding computationally intensive QR decomposition [43] or singular value decomposition (SVD). As a result, rapid identification of the TMP in the system was achieved. However, the recursive subspace method based on a fixed forgetting factor requires the size of the forgetting factor to be manually set according to prior knowledge, which lacks adaptability. Therefore, in this study, an adaptive updating method for the VFF, based on the changing speed of time-varying parameters in structures and determined by the spatial Euclidean distance of identified adjacent frequencies, was proposed to enhance the tracking efficiency of time-varying parameters in structures. Subsequently, an unbiased updating form of the Hankel matrix under the influence of the derived VFF was deduced, and the orthogonalized subspace tracking algorithm was applied to achieve recursive updating of the generalized observability matrix. This enabled the real-time estimation of modal parameters through the solution of system matrices during the identification process.

### 2.1. Problem Formulation of the ARS Method

Considering a structure with n degrees of freedom subjected to seismic loading, the discrete state–space equation at time instant t=kΔt can be formulated as:(1)$$\left\{\begin{array}{c} z_{k+1}=A_{k}z_{k}+B_{k}u_{k}\;\;\;\;\;\\ y_{k}=C_{k}z_{k}+D_{k}u_{k}+v_{k}\; \end{array}\right.$$
where $z_{k}$ is the state vector composed of the displacement and velocity at time instant t=kΔt; $u_{k}\in R^{m}$ is the input vector with m excitations; $y_{k}\in R^{p}$ is the output vector with p measurement; $A_{k}$, $B_{k}$, $C_{k}$, and $D_{k}$ are the discretized system, input, output, and feedthrough matrices, respectively; and $v_{k}$ indicates the measurement noise.

Building upon the on-line acquisition of input–output response sequences $\left\{u_{i},y_{i}\right\}\left(i=0,1,2,\cdots k\right)$, an adaptive Hankel data matrix construction mechanism with a forgetting factor was introduced. Through the integration of subspace tracking techniques, this approach achieved on-line recursive estimation of the matrix while meeting the precision and convergence requirements of the recursive algorithm. Ultimately, this enabled the on-line tracking of modal parameters at each time instant.

### 2.2. Construction of the Hankel Matrix with Forgetting Factors

Assume that the initial moment to be $t_{0}=0$ and the current moment to be t=kΔt. The input–output response sequence is $\left\{u_{i},y_{i}\right\}\left(i=0,1,2,\cdots k\right)$, $u_{i}$ is the input, and $y_{i}$ is the output. Thus, the following Hankel matrices with forgetting factors were established:(2)$$U_{0,i,k}^{\lambda_{0}}=\left[\begin{array}{cccc} \lambda_{0}^{k}u_{0} & \lambda_{0}^{k-1}u_{1} & \cdots & \lambda_{0}^{i-1}u_{k-i+1}\\ \lambda_{0}^{k-1}u_{1} & \lambda_{0}^{k-2}u_{2} & \cdots & \lambda_{0}^{i-2}u_{k-i+2}\\ \vdots & \vdots & \ddots & \vdots \\ \lambda_{0}^{k-i+1}u_{i-1} & \lambda_{0}^{k-i}u_{i-1} & \cdots & u_{k} \end{array}\right]$$
(3)$$Y_{0,i,k}^{\lambda_{0}}=\left[\begin{array}{cccc} \lambda_{0}^{k}y_{0} & \lambda_{0}^{k-1}y_{1} & \cdots & \lambda_{0}^{i-1}y_{k-i+1}\\ \lambda_{0}^{k-1}y_{1} & \lambda_{0}^{k-2}y_{2} & \cdots & \lambda_{0}^{i-2}y_{k-i+2}\\ \vdots & \vdots & \ddots & \vdots \\ \lambda_{0}^{k-i+1}y_{i-1} & \lambda_{0}^{k-i}y_{i-1} & \cdots & y_{k} \end{array}\right]$$
(4)$$V_{0,i,k}^{\lambda_{0}}=\left[\begin{array}{cccc} \lambda_{0}^{k}v_{0} & \lambda_{0}^{k-1}v_{1} & \cdots & \lambda_{0}^{i-1}v_{k-i+1}\\ \lambda_{0}^{k-1}v_{1} & \lambda_{0}^{k-2}v_{2} & \cdots & \lambda_{0}^{i-2}v_{k-i+2}\\ \vdots & \vdots & \ddots & \vdots \\ \lambda_{0}^{k-i+1}v_{i-1} & \lambda_{0}^{k-i}v_{i-1} & \cdots & v_{k} \end{array}\right]$$
where $\lambda_{0}$ is the weighted forgetting factor, $0<\lambda_{0}<1$, $U_{0,i,k}^{\lambda_{0}}$ is the input Hankel matrix with a weighted forgetting factor, $Y_{0,i,k}^{\lambda_{0}}$ is the output Hankel matrix with a weighted forgetting factor, $V_{0,i,k}^{\lambda_{0}}$ is the noise Hankel matrix with a weighted forgetting factor, and $\left\{v_{i}\right\}\left(i=0,1,2,\cdots k\right)$ is the noise sequence in the measuring process of the output response.

In the context of time-varying structural systems, the construction of a Hankel matrix incorporating forgetting factors, as depicted in Equations (2)–(4), resulted in a gradual reduction in the impact of previous input–output data as new information was incorporated during the identification process. This progressive decrease in weight ensured a more accurate depiction of the current system characteristics in the identification of structural modal parameters.

The established Hankel matrices $U_{0,i,k}^{\lambda_{0}}$ and $Y_{0,i,k}^{\lambda_{0}}$ can also be expressed as follows:(5)$$U_{0,i,k}^{\lambda_{0}}=\Lambda_{m}U_{0,i,k}\Lambda$$
(6)$$Y_{0,i,k}^{\lambda_{0}}=\Lambda_{p}Y_{0,i,k}\Lambda$$
where $\Lambda_{m}=diag\left(\lambda_{0}^{i-1}I_{m},\lambda_{0}^{i-2}I_{m},\cdots ,\lambda_{0}I_{m},I_{m}\right)$ is the m th-order diagonal matrix, and $\Lambda_{p}=diag\left(\lambda_{0}^{i-1}I_{p},\lambda_{0}^{i-2}I_{p},\cdots ,\lambda_{0}I_{p},I_{p}\right)$, $\Lambda =diag\left(\lambda_{0}^{k-i+1},\lambda_{0}^{k-i},\cdots ,\lambda_{0},1\right)$, and diag are the designated diagonal matrices.

A general state vector sequence with forgetting factors was built as follows:(7)$$Z_{0,i,k}^{\lambda_{0}}=\left[\lambda_{0}^{k-i+1}z_{0}\;\;\;\lambda_{0}^{k-i}z_{1}\;\cdots \;\lambda_{0}z_{k-i}\;\;\;z_{k-i+1}\right]$$
where $z_{j}$ designates the state vector composed of displacement and velocity at time t=jΔt.

Then, the system input–output equation with forgetting factors at the initial moment was established as follows:(8)$$\Lambda_{p}Y_{0,i,k}\Lambda =\Lambda_{p}\Gamma_{i}Z_{0,i,k}\Lambda +\Lambda_{p}H_{i}\Lambda_{m}^{-1}\Lambda_{m}U_{0,i,k}\Lambda +\Lambda_{p}V_{0,i,k}\Lambda$$
where $H_{i}$ is a lower triangular Toeplitz matrix [38,44].

Equation (8) can also be written as follows:(9)$$Y_{0,i,k}^{\lambda_{0}}=\Gamma_{i}^{\lambda_{0}}Z_{0,i,k}^{\lambda_{0}}+H_{i}^{\lambda_{0}}U_{0,i,k}^{\lambda_{0}}+V_{0,i,k}^{\lambda_{0}}$$
where $\Gamma_{i}^{\lambda_{0}}=\left[\begin{array}{ccccc} \lambda_{0}^{i-1}C & \lambda_{0}^{i-2}CA & \cdots & \lambda_{0}CA^{i-2} & CA^{i-1} \end{array}\right]$ is the generalized observable matrix with forgetting factors, and $H_{i}^{\lambda_{0}}=\Lambda_{p}H_{i}\Lambda_{m}^{-1}$ is the lower triangular Toeplitz matrix.

Therefore, the estimation of matrices $\hat{A}$, $\hat{B}$, $\hat{C}$, and $\hat{D}$ can be calculated by matrices $\Gamma_{i}^{\lambda_{0}}$ and $H_{i}^{\lambda_{0}}$ [33,45].

Since $V_{0,i,k}$ was the zero-mean random white noise, when obtaining a sufficient amount of input–output data, according to the Kalman principle, Equation (9) can be simplified as follows:(10)$$Y_{0,i,k}^{\lambda_{0}}=\Gamma_{i}^{\lambda_{0}}Z_{0,i,k}^{\lambda_{0}}+H_{i}^{\lambda_{0}}U_{0,i,k}^{\lambda_{0}}$$

To estimate $\Gamma_{i}^{\lambda_{0}}$ in Equation (9), the following least squares estimation problem was built:(11)$$\underset{H_{i}}{\min}\left\|Y_{0,i,k}^{\lambda_{0}}-H_{i}^{\lambda_{0}}U_{0,i,k}^{\lambda_{0}}\right\|{\;}_{F}^{2}$$

Then, the following Hankel matrix was built, and QR decomposition was performed:(12)$$\left[\begin{array}{c} U_{0,i,k}^{\lambda_{0}}\\ Y_{0,i,k}^{\lambda_{0}} \end{array}\right]=\left[\begin{array}{cc} R_{11}\left(k\right) & 0\\ R_{21}\left(k\right) & R_{22}\left(k\right) \end{array}\right]\left[\begin{array}{c} Q_{1}\left(k\right)\\ Q_{2}\left(k\right) \end{array}\right]$$
where $R_{11}\left(k\right)$ and $R_{22}\left(k\right)$ are lower triangular matrices, and $Q_{1}\left(k\right)$ and $Q_{2}\left(k\right)$ are orthogonal matrices.

By comparing and analyzing Equations (10) and (12), based on the principle of orthogonal projection, the following equation could then be obtained [35]:(13)$$Y_{0,i,k}^{\lambda_{0}}\prod_{U_{0,i,k}^{\lambda_{0}}}^{⊥}=R_{22}\left(k\right)Q_{2}\left(k\right)=\Gamma_{i,k}^{\lambda_{0}}Z_{0,i,k}^{\lambda_{0}}\prod_{U_{0,i,k}^{\lambda_{0}}}^{⊥}$$
where $\prod_{U_{0,i,k}^{\lambda_{0}}}^{⊥}$ is the orthogonal projection matrix of $U_{0,i,k}^{\lambda_{0}}$. 

To reconstruct the generalized observable matrix $\Gamma_{i}^{\lambda_{0}}$, the SVD of $R_{22}\left(k\right)$ was performed as follows [39]:(14)$$R_{22}\left(k\right)=\left[\begin{array}{cc} U_{1} & U_{2} \end{array}\right]\left[\begin{array}{cc} \sum {}_{1} & 0\\ 0 & 0 \end{array}\right]\left[\begin{array}{c} V_{1}^{T}\\ V_{2}^{T} \end{array}\right]=U_{1}\sum {}_{1}V_{1}^{T}$$
where $U_{1}$ is a column vector, $V_{1}^{T}$ is a row vector, and $\sum {}_{1}$ is a matrix with diagonal elements as singular values.

Therefore, the matrix $\Gamma_{i}^{\lambda_{0}}$ could be obtained as follows:(15)$$\;\Gamma_{i}^{\lambda_{0}}=U_{1}{\sum {}_{1}}^{1/2}$$

### 2.3. Updating of the Generalized Observable Matrix with VFF

Assuming that the forgetting factor at t=kΔt was $\lambda_{0}$, and the forgetting factor at $t=\left(k+1\right)\Delta t$ was $\lambda_{1}$, to facilitate the recursive update of the system matrices, at any time instant t=(k+j)Δt, the following updated input–output response update vectors were built:(16)$${\tilde{\Phi }}_{u}^{\lambda_{j}}\left(k+j\right)=\left[\begin{array}{c} {\bar{\lambda }}_{j}^{i-1}u_{k-i+j+1}\\ {\bar{\lambda }}_{j}^{i-2}u_{k-i+j+2}\\ \vdots \\ {\bar{\lambda }}_{j}^{1}u_{k+j-1}\\ u_{k+j} \end{array}\right],{\tilde{\Phi }}_{y}^{\lambda }\left(k+j\right)=\left[\begin{array}{c} {\bar{\lambda }}_{j}^{i-1}y_{k-i+j+1}\\ {\bar{\lambda }}_{j}^{i-2}y_{k-i+j+2}\\ \vdots \\ {\bar{\lambda }}_{j}^{1}y_{k+j-1}\\ y_{k+j} \end{array}\right]$$
where ${\bar{\lambda }}_{j}^{i-1}$ is the product of the first i−1 terms in the VFF sequence $\underset{j\;\mathrm{item}}{\underset{︸}{\lambda_{j}\lambda_{j-1}\cdots \lambda_{1}}}\underset{i-1\;\mathrm{item}}{\underset{︸}{\lambda_{0}\lambda_{0}\cdots \lambda_{0}}}$, and ${\bar{\lambda }}_{j}^{i-2}$ is the product of the first i−2 terms in the corresponding sequence.

At time t=(k+j)Δt, the updated Hankel matrix was given as follows:(17)$$\begin{array}{l} \left[\begin{array}{c} U_{0,i,k+j}^{\lambda_{j}}\\ Y_{0,i,k+j}^{\lambda_{j}} \end{array}\right]=\left[\begin{array}{cc} \lambda_{j}U_{0,i,k+j-1}^{\lambda_{j-1}} & {\tilde{\Phi }}_{u}^{\lambda_{j}}\left(k+j\right)\\ \lambda_{j}Y_{0,i,k+j-1}^{\lambda_{j-1}} & {\tilde{\Phi }}_{y}^{\lambda_{j}}\left(k+j\right) \end{array}\right]=R\left(k+j\right)Q\left(k+j\right)\\ =\left[\begin{array}{ccc} \lambda R_{11}\left(k+j-1\right) & 0 & {\tilde{\Phi }}_{u}^{\lambda_{j}}\left(k+j\right)\\ \lambda R_{12}\left(k+j-1\right) & \lambda R_{22}\left(k+j-1\right) & {\tilde{\Phi }}_{y}^{\lambda_{j}}\left(k+j\right) \end{array}\right]\left[\begin{array}{cc} Q_{1}\left(k+j\right) & 0\\ Q_{2}\left(k+j\right) & 0\\ 0 & 1 \end{array}\right]\\ =\left[\begin{array}{ccc} R_{11}\left(k+j\right) & 0 & 0\\ R_{21}\left(k+j\right) & \lambda R_{22}\left(k+j-1\right) & {\tilde{\Phi }}^{\lambda_{j}}\left(k+j\right) \end{array}\right]\left[\begin{array}{c} Q_{1}\left(k+j\right)\\ Q_{2}\left(k+j\right) \end{array}\right]\\ =\left[\begin{array}{cc} R_{11}\left(k+j\right) & 0\\ R_{21}\left(k+j\right) & R_{22}\left(k+j\right) \end{array}\right]Q\left(k+j\right) \end{array}$$

$R_{22}\left(k+j\right)$ in Equation (17) satisfied the following equation:(18)$$R_{22}\left(k+j\right)=\left[\begin{array}{cc} \lambda_{j}R_{22}\left(k+j-1\right) & {\tilde{\Phi }}^{\lambda_{j}}\left(k+j\right) \end{array}\right]$$
where $R_{22}\left(k+j\right)$ is a lower triangular matrix at time instant t=(k+j)Δt. 

As shown in Equation (15), after obtaining the estimate of $\Gamma_{i}^{\lambda_{0}}$, the following unconstrained optimization problem was built to achieve the recursive update of matrix $\Gamma_{i}^{\lambda_{j}}$ and reduce the computational burden introduced by the SVD decomposition [46]:(19)$$\begin{array}{cc} J\left(\omega \right) & =E\left({\left\|z-\omega \omega^{H}z\right\|}^{2}\right)\\ & =Tr\left(\Xi \right)-2Tr\left(\omega^{H}\Xi \omega \right)+Tr\left(\omega^{H}\Xi \omega \omega^{H}\omega \right) \end{array}$$
where $J\left(\omega \right)$ is the unconstrained optimization function, ω is a weight matrix, and $\Xi =E\left(zz^{H}\right)$ is a correlation matrix.

There are various recursive update algorithms for matrix ω. In this paper, the orthogonalized projection approximation subspace tracking (OPAST) algorithm was employed to achieve the recursive computation of matrix ω, and the updated estimation of the generalized observability matrix ${\hat{\Gamma }}_{i}^{\lambda_{j}}$ at time t=(k+j)Δt can be obtained [47,48].

### 2.4. Updating of System Matrix

After obtaining the recursive estimation of ${\hat{\Gamma }}_{i}^{\lambda_{j}}$, the matrix estimations $\hat{A}\left(k+j\right)$ and $\hat{C}\left(k+j\right)$ can be obtained by the following formulas:(20)$$\hat{C}\left(k+j\right)=\lambda_{j}^{1-i}{\hat{\Gamma }}_{i}^{\lambda_{j}}\left(1:p,1:2n\right)$$
(21)$$\hat{A}\left(k+j\right)={\left({\hat{\Gamma }}_{i,k+j,1}^{\lambda }\right)}^{†}\cdot {\bar{\Lambda }}_{j}\cdot {\hat{\Gamma }}_{i,k+j,2}^{\lambda }$$
where ${\hat{\Gamma }}_{i,k+j,1}^{\lambda }={\hat{\Gamma }}_{i,k+j}^{\lambda }\left(1:\left(i-1\right)p,1:2n\right)$, ${\hat{\Gamma }}_{i,k+j,2}^{\lambda }={\hat{\Gamma }}_{i,k+j}^{\lambda }\left(p+1:pi,1:2n\right)$, ${\bar{\Lambda }}_{j}=diag\left({\bar{\lambda }}_{j}^{i-1}I_{p},{\bar{\lambda }}_{j}^{i-2}I_{p},\cdots ,{\bar{\lambda }}_{j}^{1}I_{p}\right)\cdot diag\left({\bar{\lambda }}_{j}^{i-2}I_{p},\cdots ,{\bar{\lambda }}_{j}^{1}I_{p},I_{p}\right)$, and ${\hat{\Gamma }}_{i,k+j}^{\lambda }\left(1:p,1:n\right)$ are the matrix components of rows 1 to p and columns 1 to 2n of matrix ${\hat{\Gamma }}_{i,k+j}^{\lambda }$, and the other matrices are similar.

At time t=(k+j)Δt, the input–output equation of the system can be written as:(22)$$Y_{0,i,k+j}^{\lambda_{j}}=\Gamma_{i}^{\lambda_{j}}Z_{0,i,k+j}^{\lambda_{j}}+H_{i}^{\lambda_{j}}U_{0,i,k+j}^{\lambda_{j}}+V_{0,i,k+j}^{\lambda_{j}}$$

The estimation matrices $\hat{B}\left(k+j\right)$ and $\hat{D}\left(k+j\right)$ were included in the estimation matrix ${\hat{H}}_{i}^{\lambda_{j}}$, which can be obtained by eliminating ${\hat{\Gamma }}_{i}^{\lambda_{j}}Z_{0,i,k+j}^{\lambda_{j}}$ through Equation (22) through normal matrix transformations, as ${\hat{\Gamma }}_{i}^{\lambda_{j}}$ and $Z_{0,i,k+j}^{\lambda_{j}}$ can be obtained from the aforementioned steps in the paper.

### 2.5. Adaptive Updating of the VFF and TMP

The selection of forgetting factors at each moment should be determined based on the specific changes in structural parameters. When structural parameters undergo minimal changes, opting for a larger forgetting factor is recommended to mitigate the decay rate of historical data weights. Conversely, in the case of significant structural parameter variations, selecting a smaller forgetting factor is advisable for amplifying the decay rate of historical data weights. This ensures that the current input–output data carry greater weight, thereby enhancing the tracking accuracy of the identification algorithm for time-varying structural parameters.

Throughout the calculation process, the modal parameters of the structure undergo recursive updates and estimates. Designing the use of the spatial Euclidean distance between the identified adjacent modal frequencies facilitates the determination of the changing speed of structural parameters. Timely updating of the forgetting factor improves the calculation accuracy and efficiency. However, frequent updates to the forgetting factor may impact the identification accuracy of the recursive algorithm. Hence, it was essential to judiciously reduce the update rate of the VFF to enhance algorithm stability. Therefore, a method of updating forgetting factors at fixed time intervals is proposed to meet the aforementioned requirements. For a designated fixed time interval, denoted as $\tilde{p}\Delta t$, $\tilde{p}$ is a fixed step size. The disparity between the maximum and minimum values of identified frequencies within the fixed time period $\left[k\Delta t+\left(j-1\right)\tilde{p}\Delta t,k\Delta t+\left(j-1\right)\tilde{p}\Delta t\right]$ is used to assess the speed of parameter changes. The formulas for the recursive updating of the VFF and the Euclidean distance between the identified adjacent modal frequencies are given:(23)$$d\left(k+(j-1)\tilde{p},k+j\tilde{p}\right)=\sqrt{\sum_{r=1}^{n}{\left({\tilde{\omega }}_{r}^{\max}-{\tilde{\omega }}_{r}^{\min}\right)}^{2}}$$
(24)$$\lambda_{j}=\max\left(\lambda_{\min},1-c\times d\left(k+(j-1)\tilde{p},k+j\tilde{p}\right)\right)$$
where $\tilde{p}$ and c are the fixed step size and weighting factor, respectively, and their values are determined empirically; ${\tilde{\omega }}_{r}^{\max}$ and ${\tilde{\omega }}_{r}^{\min}$ are the maximum and minimum values of the identified r th frequencies within the specified time period; and $\lambda_{\min}$ is the lower limit of the forgetting factor.

Figure 1 shows the flowchart of the proposed ARS method.

## 3. Numerical Simulation Validation Example

### 3.1. Dynamic Simulation of a Concrete Arch Dam

To validate the accuracy and effectiveness of the proposed ARS method for the modal parameter recursive estimation of arch dams, a numerical simulation of a concrete arch dam under seismic loading was conducted by using the finite element analysis software ABAQUS. The concrete arch dam serves to generate and prevent floods, with a normal reservoir level set at 1880 m. The dam crest elevation and the lowest dam foundation elevation are 1885.0 m and 1580.0 m, respectively. The maximum dam height, dam crest width, and dam bottom thickness are 305.0 m, 16.0 m, and 63.0 m, respectively. Upon calculation, it was determined that extending the model 3.0 times the dam height in the upstream, downstream, left, right, and foundation directions met the precision requirements. Figure 2 shows the finite element model of the arch dam. The numerical simulation was simplified, but transverse joints were not considered. Meshing was performed using the eight-node hexahedron C3D8R and six-node pentahedron elements, resulting in a total of 142,316 nodes and 132,160 elements, with 4696 elements in the dam body.

The dynamic elastic moduli of the dam body and foundation were increased by 50% compared to their static elastic moduli [48], which were 36.9 GPa and 23.0 GPa, respectively. The Poisson ratios of the dam body and the foundation were 0.167 and 0.2, respectively. Rayleigh damping $E=a_{0}M+a_{1}K$ was used for the dam body, in which $a_{0}=0.434$ and $a_{1}=3.21\times 10^{-3}$. The damping coefficients $a_{0}$ and $a_{1}$ were obtained from the following two equations:(25)$$a_{0}=\frac{2\omega_{1}\omega_{2}(\omega_{2}\xi_{2}-\omega_{1}\xi_{1})}{\omega_{2}{}^{2}-\omega_{1}{}^{2}},\;a_{1}=\frac{2(\omega_{2}\xi_{2}-\omega_{1}\xi_{1})}{\omega_{2}{}^{2}-\omega_{1}{}^{2}}$$
where $\omega_{1}$ is the first-order natural circular frequency of the arch dam, $\omega_{2}$ is the fifth-order natural circular frequency of the arch dam, and $\xi_{1}$ and $\xi_{2}$ are the damping ratios corresponding to $\omega_{1}$ and $\omega_{2}$, respectively.

To simulate damage in a concrete arch dam, the CDP model was employed as the constitutive model for the dam body. The CDP model was constructed by integrating plastic mechanics with damage mechanics and has been incorporated into the computational analysis framework of the software ABAQUS [49,50]. This model utilizes the yield criteria from plastic mechanics to determine whether concrete has entered a state of damage. If the material is in a damaged state, the flow rule is applied to calculate the post damage inelastic strain. Subsequently, the stiffness degradation of the material after damage is calculated based on the damage evolution curve. By establishing a relationship between inelastic strain and the damage factor, the model facilitates the simulation of various degrees of structural damage. The parameters of the CDP model include the expansion angle, eccentricity ratio, and viscosity parameter, with values of 30°, 0.1, and 0.0005, respectively. A uniform failure pattern in all directions for the elements was assumed by this model, contributing to a comprehensive representation of damage in the simulation. The damage state can be intuitively observed through the obtained tensile damage factor and compressive damage factor in the CDP model. Figure 3 and Figure 4 show the relationship curves of the CDP constituent model.

Seismic loading was applied to excite the low-order frequencies of the arch dam. The designed dynamic load exhibited an acceleration peak value of 0.35 g and a duration of 30.0 s and was applied along the direction of the river. The time-history curve and Fourier spectrum curve of the horizontal seismic input acceleration are illustrated in Figure 5. Structural calculations were performed using the large-scale finite element software ABAQUS, which incorporates a compiled subroutine for the additional mass method to account for the interaction between the water and the dam. The data collection duration was 30 s, and the measuring points are shown in Figure 6. Table 1 provides the elevations of the measuring points, and a sampling frequency of 100 Hz was used.

### 3.2. Simulation Results Analysis of the Concrete Arch Dam Model

Figure 7 displays the measured acceleration responses of the arch dam, and Figure 8 displays the tensile damage of the dam body at t=30.0  s.

Varying degrees of tensile damage were observed in the dam heel, upper arch crown, and upper right dam shoulder of the arch dam, with the most severe stretching damage being experienced at the dam heel position, where a maximum damage factor DAMAGET of 1.276 is reached, while the damage to other parts was relatively minor [51]. These damaged locations also appeared in multiple finite element simulations and shaking table model tests of arch dams [10,11,52], confirming the validity of the numerical simulation results presented in this study.

To comprehend the temporal progression of the tensile damage factor DAMAGET within the dam structure, nodes b1, b2, b3, b4, and b5, depicted in Figure 8 and situated at diverse locations of the dam body, were chosen for analysis.

Figure 9 displays the progression of the DAMAGET factor over time in nodes b1, b2, b3, b4, and b5, providing a visual representation of the development of tensile damage over time. As shown in Figure 9a, the positions of nodes b1, b2, and b3 began to show damage during the time period *t* = 9.47–9.58 s, with a significant increase in damage factors observed during the time period *t* = 14.25–14.35 s at each point. In addition, the DAMAGE factors for nodes b1 and b2 both experienced slight increases during the time period *t* = 10.72–10.76 s, and the DAMAGET factor for nodes b2 and b3 showed minor increases during the time period *t* = 22.24–22.32 s. The final damage factors for nodes b1, b2, and b3 were 1.274, 0.937, and 0.986 respectively. Figure 8a and Figure 9a showed the damage characteristics and development of tensile damage at the heel of the dam.

As shown in Figure 9b, the position of node b4 began to show damage during the time period *t* = 11.47–11.49 s, and the damage in this position developed in the periods of *t* = 14.59–14.63 s and *t* = 15.6–15.62 s. In addition, the position of node b5 began to show damage only during the time period *t* = 20.6–20.62 s. Overall, the damage observed at the upper arch crown and upper right dam shoulder positions is relatively minor. The damage at the dam heel position was greater and more pronounced.

### 3.3. TMP Identification

#### 3.3.1. Time-Invariant Modal Parameter Identification

As depicted in Figure 9, there was no damage in the concrete arch dam before *t* = 8.84 s, and the structure remained time-invariant during this phase. The subspace identification method was employed to identify the time-invariant modal parameters of the structure during this phase. Considering the low frequencies and dense modes of the concrete arch dam, the measured dynamic response from *t* = 0–6.0 s was selected as the sample for computational analysis.

Due to a sampling frequency of 100 Hz, the system order was assumed to range from 2 to 100, with the frequency tolerance, damping ratio tolerance, and mode shape tolerance set at 1%, 5%, and 2%, respectively. The acceleration responses at measuring points 2, 4, and 5 were analyzed. The stability diagram in Figure 10 includes stable points of the model order and the power spectral density curves of the structural response. The horizontal axis represents the frequency, the left vertical axis indicates the model order, and the right vertical axis shows the power spectral density of the output response. Setting the number of stable points to 20, a mode is considered to be stable when the number of stable points is greater than 20, and the corresponding frequency is considered an effective structural frequency identification value. As observed in Figure 10, four orders of frequencies of the structural system were effectively identified. When the model order was greater than 90, the identification values of the various order frequencies were relatively close. When the model order was 90, the frequencies for the first four modes were 0.910, 1.166, 1.871, and 2.161 Hz, respectively.

Taking the model order as 90, the corresponding frequency and damping ratio for this order are calculated. The relative error of the identified structural modal parameters was defined as:(26)$$\delta =\left|\rho_{id}-\rho_{true}\right|/\rho_{true}\times 100\%$$
where $\rho_{id}$ is the identified frequency or damping ratio, and $\rho_{true}$ is the true value of the frequency or damping ratio.

Table 2 lists the identified values of the frequencies and the actual values calculated via finite element analysis. The identification relative error of the arch dam structure frequencies is relatively low, which also demonstrates the high accuracy of the N4SID subspace identification method for time-invariant structures.

#### 3.3.2. Modal Parameter Recursive Estimation

The modal frequencies of large concrete arch dam structures were relatively dense, with a first-order frequency even below 2.0. When applying the proposed ARS method in this study to the modal parameter recursive estimation of the concrete arch dam, calculations had determined that effective identification of the modal parameters of the structure is achieved when the values of k and i in Equations (2) and (3) are 520 and 90, respectively. When *k* is less than 520 or even smaller, the modal parameters of the arch dam cannot be effectively identified. When i is less than 90, and especially below 80, the stability and robustness of the identified frequencies decrease. Clearly, for different structures or systems, the values of *k* and those required for effective modal parameter estimation using the ARS method will vary.

In addition, the lower limit of the forgetting factor was set to 0.95, the weighting factor λ was set to 0.1, and the fixed time interval was set to 0.1 s. Since k was set to 520, requiring a dynamic response sequence length of 5.20 s, the analysis period for time-varying parameters ranged from 5.2 s to 25.0 s.

Figure 11 illustrates the evolution of the VFF during the identification process. At *t* = 9.5 s and *t* = 14.3 s, a sudden increase in structural damage results in significant differences in the frequency identification values at consecutive moments. Consequently, the forgetting factor rapidly drops to 0.95 before gradually recovering. Once the frequency identification values at consecutive moments stabilize and become nearly identical, the forgetting factor returns to 1.0. The variations in the forgetting factor also indirectly indicate the rate of change in the structure’s frequency identification values.

Figure 12, Figure 13, Figure 14 and Figure 15 depict the modal parameter recursive estimation values of the frequencies and damping ratios for the first four modes of the arch dam structure. The modal parameter identification values were relatively stable both before any damage occurred and after the development of stable damage. In the undamaged state, the identification values for the first four frequencies remained at approximately 0.910, 1.166, 1.871, and 2.161 Hz, while the damping ratio identification values for the first four modes were approximately 4.44%, 4.28%, 5.42%, and 5.56%, respectively. After *t* = 19.5 s, the modal parameter identification values tended to stabilize. The first four frequencies were approximately 0.895, 1.134, 1.842, and 2.134 Hz, and the damping ratios for the first four modes were approximately 4.98%, 4.91%, 6.61%, and 6.85%, respectively.

Throughout the entire identification process, significant changes in modal parameter identification values became apparent during the time period *t* = 9.47–9.58 s. During this time period, damage began to occur at the dam heel, with the DAMAGET factors at nodes b1 and b3 increasing from 0 to 0.423 and 0.321, respectively. Subsequently, the first four frequencies exhibited a gradual overall decrease, while the first four damping ratio identification values showed a general upward trend. This corresponded to the enlargement of the damaged area and an increase in the severity of damage, leading to a decrease in structural frequency and an increase in the damping ratio for the arch dam structure. During the time period *t* = 14.25–14.35 s, the identified frequencies of the four orders exhibited greater oscillation compared to other time periods, attributed to the rapid increase in DAMAGET factors at the dam heel and upper arch positions during this period. According to Figure 8, Figure 9, Figure 10 and Figure 11, during the time period *t* = 14.25–19.5 s, the decrease in the identified frequency amplitudes of the four orders and the increase in the damping ratios of the four orders were greater than those during the time period *t*= 9.47–14.24 because the increase in damage to the dam body during the time interval *t* = 14.25–14.35 was more severe. Therefore, the changes in the modal parameters align with the expansion and intensification of structural damage. However, the identified modal parameters were not sensitive to minor damage occurring during the time period *t* = 20.6–20.62 s.

Combining the results of the time-dependent damage progression in the arch dam model with the recursive values of frequency and damping ratio obtained using the ARS method, it can be observed that the ARS method cannot immediately and accurately identify the frequencies and damping ratios of all modes after the occurrence of structural damage. In summary, the identified modal parameters of the arch dam structure are based on the structural dynamic response data, which requires a certain duration of data. As previously mentioned, calculations show that when k is less than 520, the modal parameters of the structure cannot be effectively identified. This also indicates that using the ARS method requires nearly 4–5 s of recursive identification to approximate the current accurate frequencies and damping ratios of the arch dam.

In addition, the recursive subspace method based on a fixed forgetting factor requires the size of the forgetting factor to be manually set according to prior knowledge, which lacks adaptability. Compared with the RS method based on a fixed forgetting factor, the proposed ARS method exhibits better time-varying tracking characteristics.

On the other hand, as with other methods, noise will also affect the effectiveness of the proposed ARS method for recursive identification of modal parameters. Clearly, the greater the noise, the greater the impact on the modal parameter identification results; the smaller the noise, the lesser the impact. It is expected that the ARS method proposed in this paper has a certain level of robustness to noise, similar to other RS methods, especially for low noise levels. Similarly, it is expected that for the dynamic response of concrete arch dam structures, improving the signal-to-noise ratio through denoising and applying the proposed ARS method can effectively achieve the modal parameters’ recursive estimation of the arch dams. Future works will explore and discuss the impact of noise on the proposed ARS method.

## 4. Conclusions

Under seismic loading, whether the state of a concrete dam structure changes is a focal point of engineering safety concerns. Modal parameters, as crucial indicators for evaluating structural dynamic performance with clear physical significance, naturally become the primary means for assessing whether there is a change in the structural dynamic performance of a concrete dam. This study explored the use of the ARS identification method for the modal parameter recursive estimation of concrete arch dams. The main contents of the paper are as follows:(1)An ARS method with VFF was proposed to achieve the modal parameter recursive estimation of concrete arch dams. The VFF was adaptively updated by using the spatial Euclidean distance information of the identified adjacent modal frequency values. The introduction of the variable forgetting factor (VFF) allowed for adaptive adjustments based on the spatial Euclidean distance of the structural modal frequencies between adjacent time points. Compared to the constant forgetting factor and the moving window method, the adaptive recursive subspace method offered higher efficiency in tracking time-varying parameters during time-varying phases. In time-invariant phases, the value of the forgetting factor was 1.0, which resulted in lower computational complexity and higher efficiency compared to the constant forgetting factor method.(2)A numerical simulation of a concrete arch dam with a maximum dam height of 305.0 m under seismic loading was conducted, and the acceleration responses of the dam were obtained. The CDP model was employed to simulate the dam body’s constitutive relation, through which the damage development of the dam body can be obtained. Seismic loading was applied to motivate the arch dam, and substantial tensile damage was detected in the upper arch crown and dam heel, with no observable damage in either the abutment or the central part of the dam. Simultaneously, the occurrence time and development process of plastic damage in the dam body were obtained, with the maximum DAMAGET factors for the dam heel and arch crown being 1.274 and 0.164, respectively.(3)The identification results of the ARS method for the arch dam demonstrated a degree of sensitivity to structural damage. However, they are not particularly sensitive to minor damage. The four frequencies of the arch dam gradually decreased from the initial undamaged state of 0.910, 1.166, 1.871, and 2.161 Hz to final values of 0.895, 1.134, 1.842, and 2.134 Hz, respectively. Simultaneously, the four damping ratios increased from the initial undamaged state of 4.44%, 4.28%, 5.42%, and 5.56% to the final values of 4.98%, 4.91%, 6.61%, and 6.85%, respectively. The identification results of the time-varying modal parameters were generally consistent with the damage progression trend of the arch dam, thereby validating the effectiveness of the proposed ARS method.(4)The research findings of this paper hold certain reference value for revising content related to real-time structural modal parameter identification and damage diagnosis in the technical specifications for seismic monitoring of concrete arch dams. For example, according to the numerical simulation results, reinforcing seismic monitoring and damage diagnosis at the upper arch crown and dam toe positions could be considered, and the proposed method offers valuable insights and essential references for tracking the time-varying modal parameters of arch dams exposed to strong seismic effects.

## Figures and Tables

**Figure 1 sensors-24-03845-f001:**
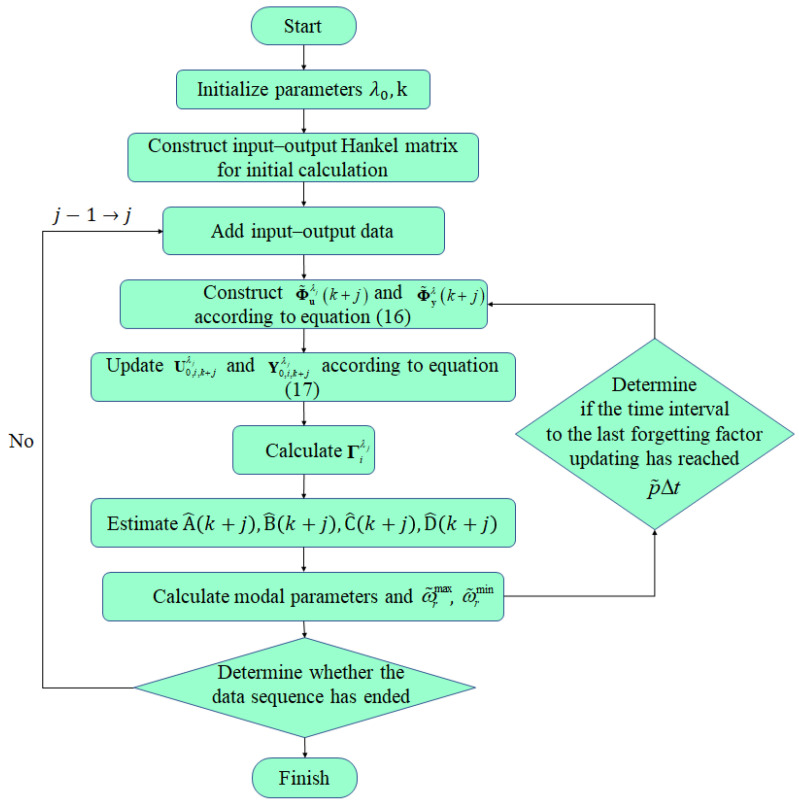
Flowchart of the proposed ARS method.

**Figure 2 sensors-24-03845-f002:**
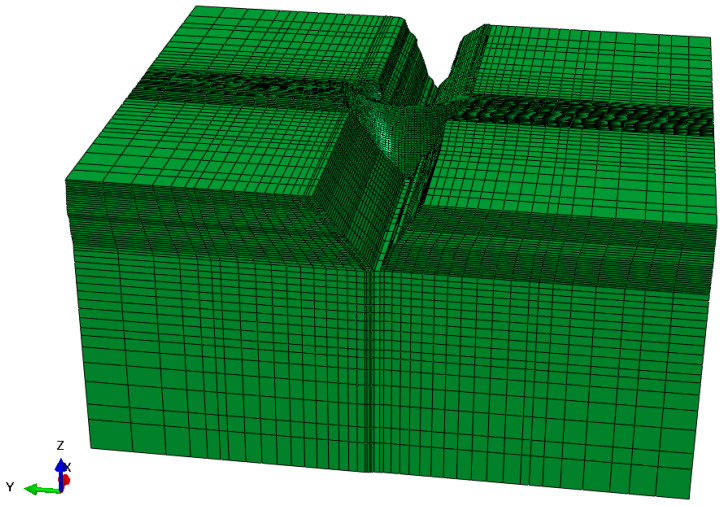
Finite element model of a concrete arch dam.

**Figure 3 sensors-24-03845-f003:**
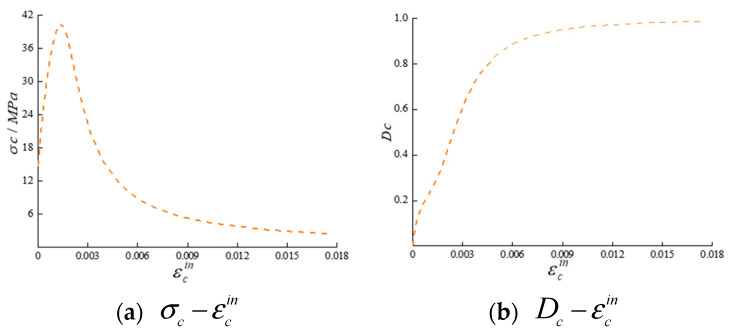
Relationships among the compressive stress $\sigma_{c}$, inelastic compressive strain $\varepsilon_{c}^{in}$, and compressive damage factor $D_{c}$.

**Figure 4 sensors-24-03845-f004:**
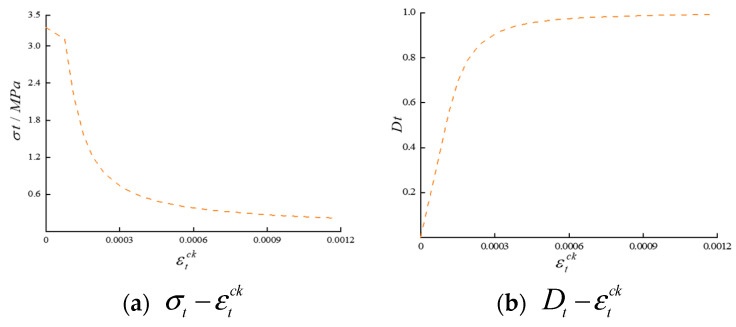
Relationships among the tensile stress $\sigma_{t}$, cracking strain $\varepsilon_{t}^{ck}$, and tensile damage factor $D_{t}$.

**Figure 5 sensors-24-03845-f005:**
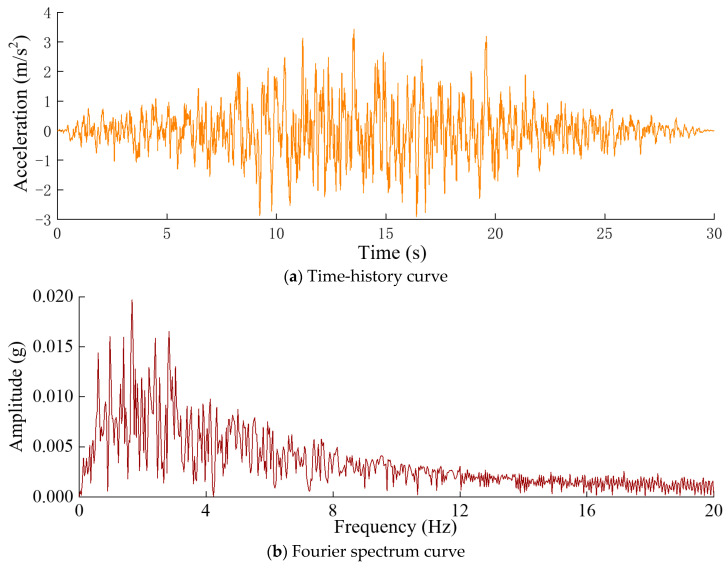
Time-history curve and Fourier spectrum curve of the horizontal seismic input on the arch dam model.

**Figure 6 sensors-24-03845-f006:**
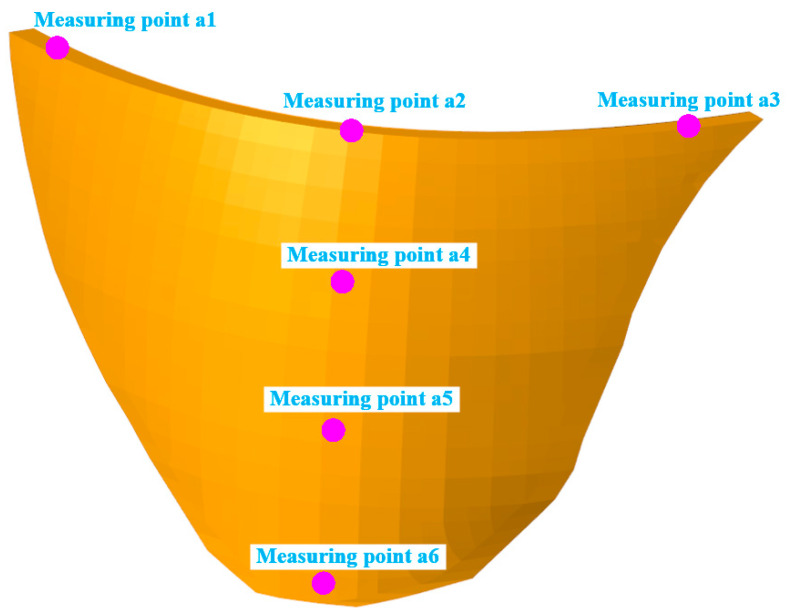
The location of the dynamic response measuring points.

**Figure 7 sensors-24-03845-f007:**
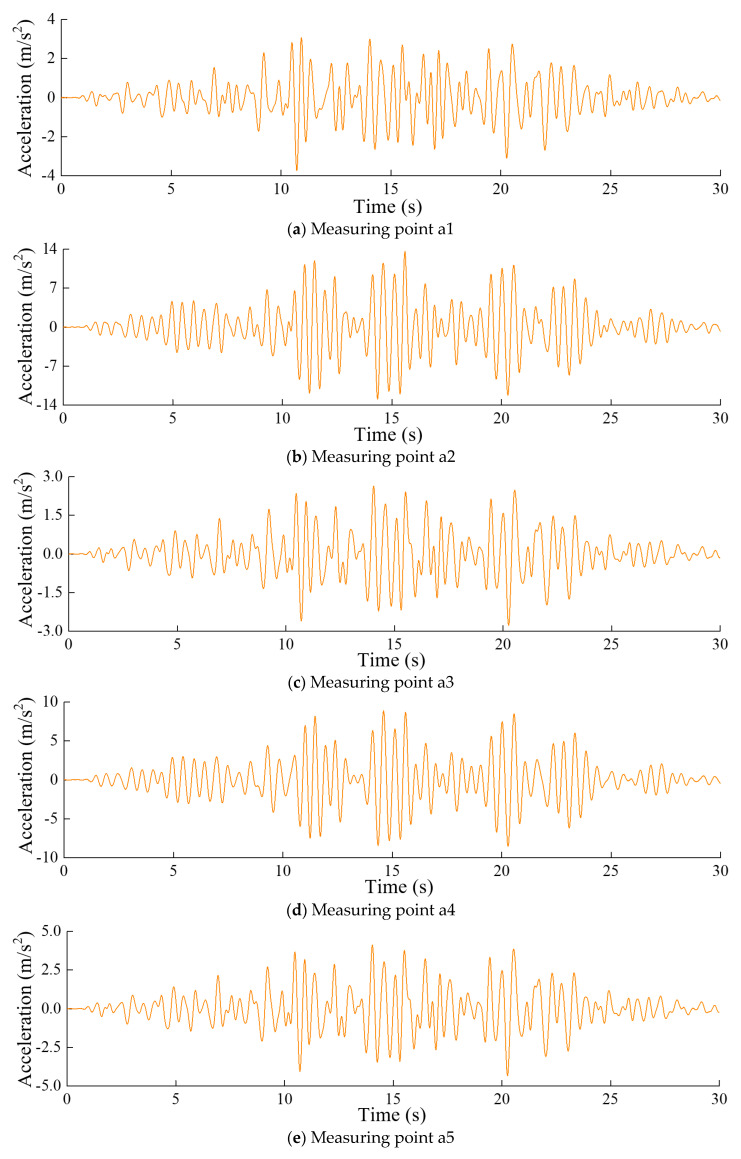
Time-history curve of the dynamic response at each measurement point.

**Figure 8 sensors-24-03845-f008:**
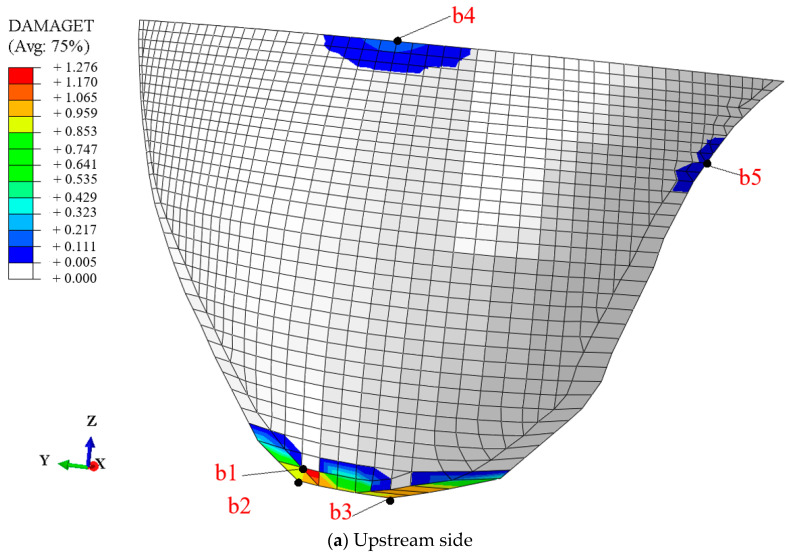
Tensile failure diagram of the dam body at t=30.0 s.

**Figure 9 sensors-24-03845-f009:**
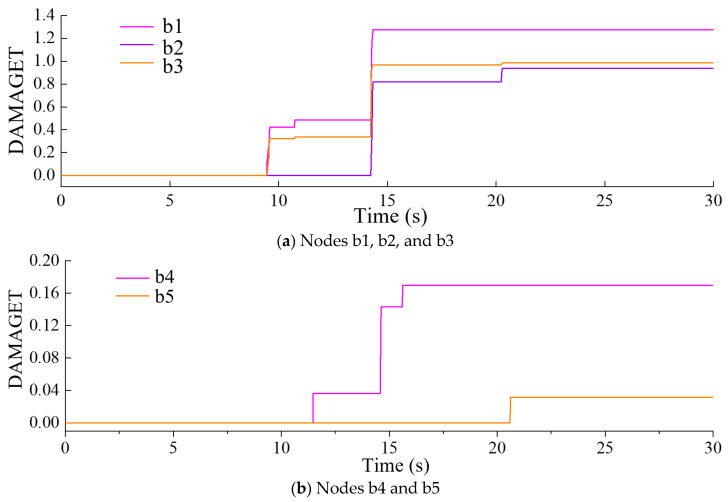
Time history of the DAMAGET factors at different nodes of the dam.

**Figure 10 sensors-24-03845-f010:**
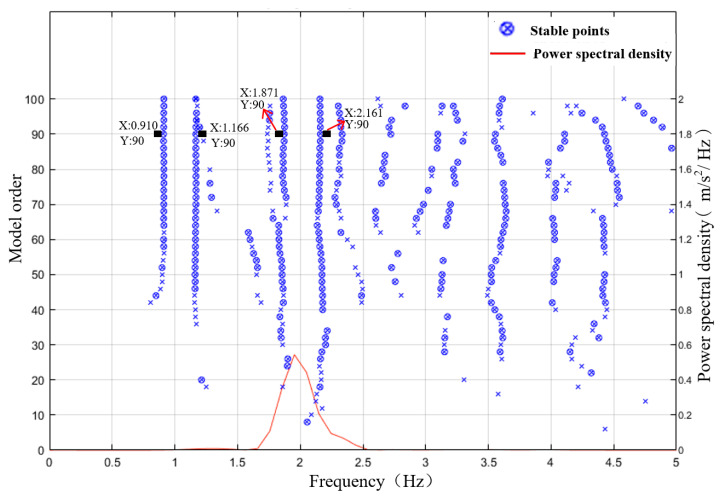
Stability diagram of the N4SID (numerical algorithms for subspace state space system identification) method.

**Figure 11 sensors-24-03845-f011:**
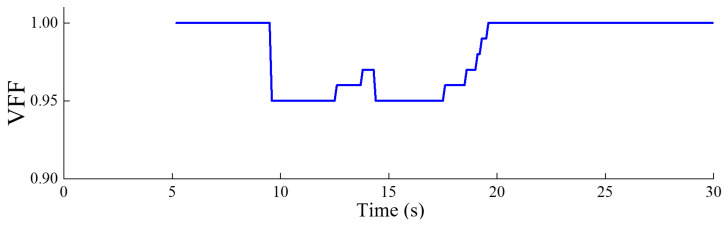
Evolution of the VFF during the identification process.

**Figure 12 sensors-24-03845-f012:**
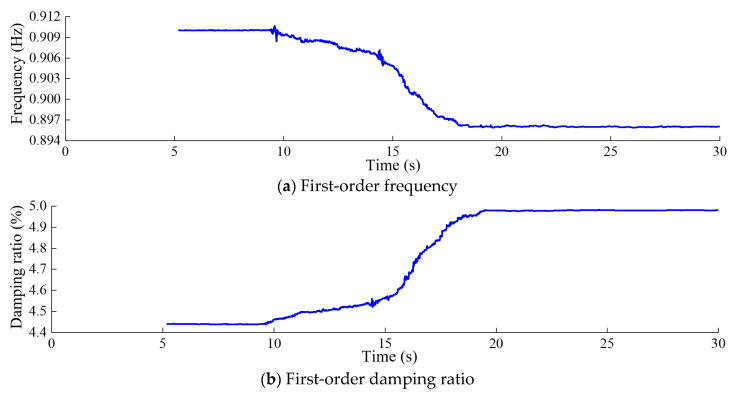
First-order modal parameter recursive estimation for the arch dam.

**Figure 13 sensors-24-03845-f013:**
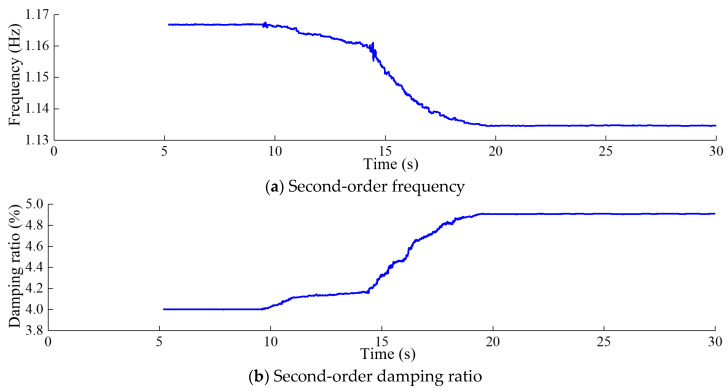
Second-order modal parameter recursive estimation for the arch dam.

**Figure 14 sensors-24-03845-f014:**
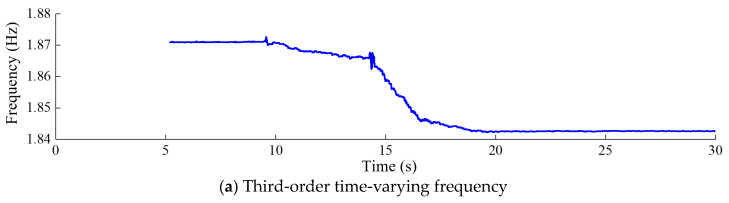
Third-order modal parameter recursive estimation for the arch dam.

**Figure 15 sensors-24-03845-f015:**
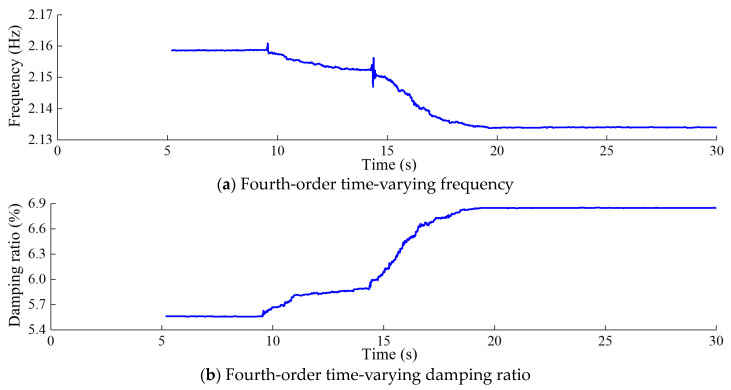
Fourth-order modal parameter recursive estimation for the arch dam.

**Table 1 sensors-24-03845-t001:** Elevation of the dynamic response collection points.

Measuring Point	Elevation (m)	Measuring Point	Elevation (m)
a1	1885	a2	1885
a3	1885	a4	1790
a5	1695	a6	1600

**Table 2 sensors-24-03845-t002:** Fourth-order frequencies identified by the N4SID method.

Order	Identified Frequency (Hz)	True Frequency (Hz)	Relative Error (%)
1	0.910	0.913	0.329
2	1.166	1.167	0.086
3	1.871	1.877	0.320
4	2.161	2.159	0.093

## Data Availability

Data are contained within the article.

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
