# Peer review of "Modal Parameter Recursive Estimation of Concrete Arch Dams under Seismic Loading Using an Adaptive Recursive Subspace Method"

_sensors, 2024, doi:10.3390/s24123845_

Round 1

Reviewer 1 Report (New Reviewer)

Comments and Suggestions for Authors This study focused on the crucial task of modal parameter estimation for vibration-based damage detection in concrete arch dams, particularly under severe seismic events. The research introduced an adaptive recursive subspace method with variable forgetting factors to estimate modal parameters recursively. A numerical simulation of a concrete arch dam under seismic loading was performed using the ABAQUS software, incorporating the concrete damaged plasticity (CDP) model to assess damage development. The ARS method, utilizing dynamic responses from the simulation, identified a decreasing trend in the frequencies of the dam's initial four modes and an increase in the damping ratios of these modes. The results confirmed the correlation between the identification results and damage progression, validating the effectiveness of the ARS method.   Modal parameters, including natural frequency, mode shape, and damping ratio, reflect the inherent characteristics of a structure and have clear physical significance. They are important indicators for understanding the dynamic properties of a structure. These parameters can provide crucial reference points for structural damage identification, model updating, and the optimal design of structural dynamic properties. Therefore, given the importance of concrete arch dam engineering and the potential damage that may occur under strong seismic loads, this article's focus on the modal parameters recursive identification arch dams under seismic loading holds significant engineering value and scientific importance. For the introduction section, the introduction of this study is commendable for its comprehensive and well-articulated presentation of the research background. The comprehensive discussion, particularly concerning the dynamic response monitoring , structural damage identification and modal parameter identification of concrete arch dams, is impressive. These comprehensive discussions significantly enhance the quality of the paper, making it one of its key strengths. In addition, the authors conducted a successful numerical simulation of an arch dam, the damage positions and damage development of the dam were obtained, and the damage positions were common damage locations. The modal parameters recursive identification results by the adaptive recursive subspace in Section 3.3 are considered effective, as they are consistent with the damage progression results obtained from the numerical simulation in Section 3.2 and can identify structural damage.   In summary, this paper explores an interesting topic that is well-received by readers, particularly researchers and professionals in the field of structural health monitoring of concrete dams. The paper is clearly written and well-organized, with a solid structure and reliable research results. Therefore, I recommend accepting this paper, but after a careful review of the entire text, the following minor revisions must be made before its acceptance:     1. Lines 30-33: The final frequency values and damping ratio values of the the arch dam's initial four modes mentioned here are different from the corresponding values in Section 3.3.2, Figures 10 to 13 and Conclusion, while the frequency values and damping ratio values are consistent in Section 3.3.2, Figures 10 to 13, and the conclusion. By reviewing the explanation in Section 3.3.2, particularly the identification results shown in Figures 10 to 13, the final frequency values mentioned in this section should be 0.895, 1.134, 1.842, and 2.134 Hz respectively, and the final fourth-order damping ratio values should be 4.98%, 4.91%, 6.61%, and 6.85% respectively. All other descriptions regarding the identification results in the entire document are consistent, but the results mentioned in this part of the abstract are evidently incorrect and need to be rectified for consistency. Therefore, these values must be revised and should be explained.   2. Line 158: The full name of RS is missing. Although the full name of RS, which stands for recursive subspace, can be inferred by reading the entire text, it is still recommended to write out the full name for the sake of writing rigor in the paper.   3. Lines 246-249: References on conventioanl subspace methods was recommended to be included to enhance readers' understanding.   4. Lines 390-391, Please provide the basis or reference for the statement that the dynamic elastic modulus is increased by 50% compared to the static elastic modulus.   5. Line 397: This line needs to be left-aligned, just like Line 289.   6. Line 400: The references about the CDP model are recommended to be added to assist readers in understanding the interpretation of the CDP model and its parameters.   7.Lack of some latest research progress, such as “Intelligent calibration method for microscopic parameters of soil‒rock mixtures based on measured landslide accumulation morphology”   8. Lines 495-496: The formula number should be modified; it should not be (1) but (25). 

Author Response

Dear reviewer 1:

We are writing to express our sincere gratitude for your thorough review of our manuscript titled "Modal Parameters Recursive Estimation of Concrete Arch Dams Under Seismic loading Using an Adaptive Recursive Subspace Method ".

Your insightful comments and suggestions have been immensely valuable in refining the quality of our work. We would like to inform you that we have carefully considered each of your recommendations and have incorporated them into the revised version of the manuscript. Your expertise has undoubtedly contributed to enhancing the clarity, coherence, and overall strength of our research.

We truly appreciate the time and effort you dedicated to providing constructive feedback, and we believe that your input has significantly improved the overall quality of our manuscript. Your commitment to advancing scholarly work is commendable, and we are grateful for your dedication to the peer review process.

Please find attached the revised manuscript and the reponse for your perusal. We hope that the changes made align with your expectations, and we are open to any further suggestions you may have.

Once again, thank you for your invaluable contribution to our research. We look forward to hearing from you soon.

Sincerely yours,

Xinyi Zhu ,Jianchun Qiu ,Yanxin Xu ,Xingqiao Chen , Pengcheng Xu  , Xin Wu , Shaolong Guo , Jicheng Zhao , and Jiale Lin

Reviewer 2 Report (New Reviewer)

Comments and Suggestions for Authors

The paper titled "Modal parameters recursive estimation of concrete arch dams under seismic loading using an adaptive recursive subspace  method" has been reviewed. The authors conducted the modal parameters recursive estimation of arch dams, in which an adaptive recursive subspace method with variable forgetting factors was proposed. Then, a numerical simulation of an arch dam was conducted by using ABAQUS, through which the damage pattern and damage development over time were obtained. Finally, the modal parameters recursive estimation of the arch dam were conducted for the arch dam, and the identification results were consistent with the simulation results of the damage.

In my opinion, the research is both interesting and valuable, with scientifically significant and engineering application potential for concrete arch dams. The introduction section is  well-written, reflecting a deep understanding of the research field. The structure of the paper is logical and well-organized, with clear and precise expression throughout. The conclusions drawn are reliable, and the research findings offer valuable insights.

Considering the merits of this study, this paper is worthy of acceptance. However, several minor revisions should be conducted to further enhance the quality of the manuscript. The suggested modifications are outlined below:

(a) The final frequency and damping ratio values presented in the abstract (Line 30 to Line 33) are inconsistent with the identification results and should be revised.

(b) Section 2: The full name of QR and SVD ought be given. 

(c) The definitions of all variable symbols in the formulas have already been provided after each formula, so there is no need to explain them all again in Table 1 to avoid repetition and redundancy. It is recommended to either remove Table 1 or delete the explanations of the variables in the text.

(d) Line 298: The sizes of B and D appear inconsistent. The font styles for different types of variable symbols should be standardized respectively. Check the fonts of all variable symbols.

(e) Line 362 to Line 365 were similar to Line 366 to Line 369, A and V represent the fixed step size and weighting factor? C and D represented the maximum and minimum values ?  Need to check the errors here. 

(f) Section 3.2 and Section 3.3: Regarding the unit of time 𝑡 , whether it is described as 's' or 'seconds' needs to be consistent throughout the text. For example, line 452 to line 464.

Comments on the Quality of English Language

Typing and grammatical errors should be corrected including the format of some equations.

Author Response

Dear reviewer 2

We are writing to express our sincere gratitude for your thorough review of our manuscript titled "Modal Parameters Recursive Estimation of Concrete Arch Dams Under Seismic loading Using an Adaptive Recursive Subspace Method ".

Your insightful comments and suggestions have been immensely valuable in refining the quality of our work. We would like to inform you that we have carefully considered each of your recommendations and have incorporated them into the revised version of the manuscript. Your expertise has undoubtedly contributed to enhancing the clarity, coherence, and overall strength of our research.

We truly appreciate the time and effort you dedicated to providing constructive feedback, and we believe that your input has significantly improved the overall quality of our manuscript. Your commitment to advancing scholarly work is commendable, and we are grateful for your dedication to the peer review process.

Please find attached the revised manuscript and response for your perusal. We hope that the changes made align with your expectations, and we are open to any further suggestions you may have.

Once again, thank you for your invaluable contribution to our research. We look forward to hearing from you soon.

Sincerely yours,

Xinyi Zhu ,Jianchun Qiu ,Yanxin Xu ,Xingqiao Chen , Pengcheng Xu  , Xin Wu , Shaolong Guo , Jicheng Zhao , and Jiale Lin

Reviewer 3 Report (New Reviewer)

Comments and Suggestions for Authors

The paper aims to present a novel approach called the adaptive recursive subspace (ARS) method for recursively estimating the time-varying modal parameters of concrete arch dams subjected to seismic loading. It introduces variable forgetting factors (VFF) that adapt based on the changing rates in identified modal parameters, allowing for improved tracking performance. In the current manuscript, the proposed method is first introduced and derived. To verify the proposed method, a numerical simulation is conducted, and the identified results are exhibited and discussed. Finally, some conclusions are drawn at the end of the manuscript.

The authors try to provide a comprehensive theoretical derivation of the ARS method, detailing the construction of Hankel matrices with forgetting factors, the recursive updating of the generalized observability matrix, and the calculation of system matrices to obtain modal parameter estimates. Overall, the manuscript is well-organized and -presented. However, the derivation is unclear and has a lot of flaws in the current version. The paper focuses solely on numerical simulations, and experimental validation or field application using the proposed method is not provided. Additionally, while the authors discuss the method's sensitivity to minor damage, a more comprehensive analysis of its limitations or potential challenges in practical implementations could further strengthen the study. Therefore, the manuscript needs some major revisions. The following are the comments or issues existing in the current version.

1.          The authors should emphasize the novelties of their proposed method when deriving the theoretical framework. Clearly highlighting the unique contributions of this study would enhance the clarity and impact of the paper.

2.          It is important for the authors to address this concern and discuss the limitations or potential applications of their method. For example, the parameter tuning, the computational complexity, the influence of measurement noise, and the spurious modes may deteriorate the performance, which is common case in practice.

3.          The introduction section seems too long, and some materials can be simply curtailed for clarity. For example, the reviews of time-invariant methods since this manuscript focuses on time-varying methods.

4.          The introduction section should encompass a comprehensive survey of the most recent contributions within the field related to the proposed method. Currently, it primarily focuses on non-subspace methods and time-invariant methods; leaving a lot of gaps between the previous and proposed methods. For example, Weng and Loh (2011) also proposed a recursive subspace method based on QR decomposition. Kameyama et al. (2005) and Huang et al. (2022) used a bona fide recursive algorithm to achieve subspace identification rapidly, to name but a few.
https://doi.org/10.1016/j.ymssp.2011.05.013
https://doi.org/10.1016/j.automatica.2007.03.029
https://doi.org/10.3390/app122110841

5.          In Line 174, the authors mentioned the abbreviation RS without any clear definition. Please provide a full name when the terminology first appears.

6.          In Equation (1), the measured output vector, yk, is bold for a vector, but in the following derivation, it is italic for a scale. Please correct them for better understanding.

7.          In Line 283, the authors said 'depicted in equations (1)-(2)' but it should be equations (2)-(4).

8.          In section 2, n, m, p, and P aren't specified. Please clarify them for better understanding.

9.          In Equations (8) and (9), the lower triangular Toeplitz matrix, Hi, is used but not introduced. If the authors do not specify it, please provide some references. Moreover, the initial letter should be capitalized because Toeplitz is a name.

10.      Similarly, some references are needed to explain the relationship between Equations (12), (13) and (14).

11.      In Equation (18), the left-most term isn't specified. Please clarify them for better understanding.

12.      In Equation (19), it's unclear how to achieve the recursive update of the generalized observable matrix. Please clarify it in the manuscript.

13.      In Equation (19), does z have the same definition as the one in Equation (1)? Please replace it to avoid misunderstanding.

14.      In Line 326, the reviewer thinks that the PAST should stand for projection approximation subspace tracking, which Yang developed in 1993 and 1995. Therefore, the OPAST should be orthogonalized projection approximation subspace tracking.

15.      In Line 338, how can the state Hankel matrix, Z, be eliminated to estimate the lower triangular Toeplitz matrix as well as B and D?

16.      In Line 362, A and V do not represent the fixed step size and weighting factor. Similar mistakes can be found regarding C, D, and f in this paragraph.

17.      Considering that the derivation is somewhat complex, a flowchart for applying the proposed method in Section 2 is helpful.

18.      Considering the dam size (the height, crest width, and bottom thickness were 305.0 m, 16.0 m, and 63.0 m), inputting only one seismic waveform seems to be impractical. What would be the results if the seismic waveforms were different from the left-most side to the right-most side? Furthermore, which sensors in Figure 5 are taken as input in the proposed method to consider the seismic forces? Are those measurements representative subjected to the earthquake excitations?

19.      In Figure 7, are Figure 7(a) and Figure 7(b) both in the upstream side?

20.      In Section 3, the damage factor, DAMAGET, is used but not introduced. Please provide some references.

21.      The authors are encouraged to show and discuss the VFF in this seismic event. This can be used to check the mechanism shown in Section 2.5.

22.      At the same time, what is the benefit resulting from the VFF? Would a constant forgetting factor fail to identify the time-varying modal parameters? Please clarify them in the manuscript.

23.      In Line 568, what is the meaning of oscillatory nature? The modal parameters may vary over time but should not oscillate in the field application.

Comments on the Quality of English Language

Moderate editing of English language required

Author Response

Dear reviewer 3

We are writing to express our sincere gratitude for your thorough review of our manuscript titled "Modal Parameters Recursive Estimation of Concrete Arch Dams Under Seismic loading Using an Adaptive Recursive Subspace Method ".

Your insightful comments and suggestions have been immensely valuable in refining the quality of our work. We would like to inform you that we have carefully considered each of your recommendations and have incorporated them into the revised version of the manuscript. Your expertise has undoubtedly contributed to enhancing the clarity, coherence, and overall strength of our research.

We truly appreciate the time and effort you dedicated to providing constructive feedback, and we believe that your input has significantly improved the overall quality of our manuscript. Your commitment to advancing scholarly work is commendable, and we are grateful for your dedication to the peer review process.

Please find attached the revised manuscript and response for your perusal. We hope that the changes made align with your expectations, and we are open to any further suggestions you may have.

Once again, thank you for your invaluable contribution to our research. We look forward to hearing from you soon.

Sincerely yours,

Xinyi Zhu ,Jianchun Qiu ,Yanxin Xu ,Xingqiao Chen , Pengcheng Xu  , Xin Wu , Shaolong Guo , Jicheng Zhao , and Jiale Lin

Round 2

Reviewer 3 Report (New Reviewer)

Comments and Suggestions for Authors

The paper aims to present a novel approach called the adaptive recursive subspace (ARS) method for recursively estimating the time-varying modal parameters of concrete arch dams subjected to seismic loading. Noteworthy, the paper focuses solely on numerical simulations, meaning that experimental validation or field application using the proposed method is not provided. Moreover, the method's limitations or potential challenges in practical implementations are somehow missed. Last but not least, the derivation can still be improved in the current version.

After the first round of review, most of my concerns were properly addressed, but some comments or suggestions remained after viewing the revised version. Therefore, the manuscript needs some major revisions. The following are the comments or issues unsolved in the current version.

1.          The authors fail to clearly emphasize the novelties of their proposed method when deriving the theoretical framework. From the reviewer's viewpoint, the variable forgetting factor (VFF) is the only novelty, but it's really vague how it benefits identification results compared to a constant forgetting factor or a moving window solution.

2.          It is important for the authors to address this concern and discuss the limitations or potential applications of their method. For example, the parameter tuning, the computational complexity, the influence of measurement noise, and the spurious modes may deteriorate the performance, which is common case in practice. In the revised version, the authors still have no discussion on the method's limitations or potential issues.

3.          The introduction section seems too long, and some materials can be simply curtailed for clarity. Although the reviews of time-invariant methods were skipped in the first round of review, the authors added more materials, so this section was actually extended. Most importantly, not all of them are related to the improvement of the proposed method.

12.      In Equation (19), it's unclear how to achieve the recursive update of the generalized observable matrix. Please clarify it in the manuscript, according to orthogonalized projection approximation subspace tracking (OPAST) algorithm.

13.      In Equation (19), does z have the same definition as the one in Equation (1)? Please replace it to avoid misunderstanding.

15.      In Line 415 and Line 416, how can the state Hankel matrix, Z, be eliminated to estimate the lower triangular Toeplitz matrix as well as B and D?

18.      Considering the dam size (the height, crest width, and bottom thickness were 305.0 m, 16.0 m, and 63.0 m), inputting only one seismic waveform seems to be impractical. What would be the results if the seismic waveforms were different from the left-most side to the right-most side? Furthermore, which sensors in Figure 6 are taken as input in the proposed method to consider the seismic forces? Are those measurements representative subjected to the earthquake excitations?

22.      What is the benefit resulting from the VFF? Please clarify them in the manuscript. Would a constant forgetting factor fail to identify the time-varying modal parameters? Why frequent updates to the forgetting factor may impact the identification accuracy of the recursive algorithm because the forgetting factor only controls the fading rate but not the changes the data at all.

Author Response

Dear reviewer,   see attached file.   Best regards,  

Round 3

Reviewer 3 Report (New Reviewer)

Comments and Suggestions for Authors

Most of my concerns are properly addressed and the readers could now better understand the adaptive recursive subspace (ARS) method aimed to recursively estimate the time-varying modal parameters of concrete arch dams subjected to seismic loading. Although a few deficiencies still exist, for example, only numerical simulations were conducted, and the method's limitations or potential challenges aren't well-addressed, the current version responded to those comments well. Overall, the proposed method and the manuscript is well-organized and -presented.

This manuscript is a resubmission of an earlier submission. The following is a list of the peer review reports and author responses from that submission.

Round 1

Reviewer 1 Report

Comments and Suggestions for Authors

This is an interesting study. Timely identification of earthquake damage for arch dam concrete was done by analyzing the variations in structural dynamic properties measured by the dynamic response of concrete dams.

According to the iThenticate report, the similarity rate is 28% and there are many similarities in parts of the abstract and introduction and other parts of this article about the main work done in this research, please explain the difference with the article “Online structural damage state identification of concrete arch dams under dynamic loads using a recursive TVARX approach” completely in explain the introduction. Bring the innovation of this research  compare to the article “Online structural damage state identification of concrete arch dams under dynamic loads using a recursive TVARX approach” in the form of comments, in the abstract, and the introduction.

I believe rather than just reporting the results, it is important to provide a critical discussion of the findings. At the moment, the paper mostly looks like a report rather than a scientific paper. The results and discussions have few discussions based on current literature and comparisons among other papers. Therefore, a major revision is required before accepting the paper for publication. 

Other minor and major suggestions are described below:

1-      Time-varying Modal Parameters Identification of Concrete Arch Dam Using an Adaptive Recursive Subspace Method

Time-varying Modal Parameters Identification…. many of words including nouns and adjectives are listed. I suggest “Identification of Time-varying Modal Parameters for Concrete Arch Dam Using an Adaptive Recursive Subspace Method”

2-      All the verbs that express the research done by the authors should be expressed in the past tense.

 “time-varying modal parameters”

This term is repeated a lot in the text of the article, please abbreviate it as "TMP".

3-       from lines 16 to 20,  “the identification of the time-varying modal parameters of concrete arch dams”.

It has been repeated redundantly.

In this study, the identification of the time-varying modal parameters of arch dams under seismic loads is studied.”

I suggest “ In this study, the identification of the time-varying modal parameters of arch dams under seismic loads was a challenge that an adaptive recursive subspace method with variable forgetting factors used to solve it.” that the authors could change it.

4-       line 20: To validate the effectiveness of the proposed adaptive recursive subspace method, a numerical simulation of a concrete arch dam under seismic loading is conducted, and the acceleration responses of the dam are obtained.

You cannot use numerical modeling to verify the method, the question is how to validate and verify this numerical modeling, I suggest

A numerical simulation of a concrete arch dam under seismic loading was conducted, and the acceleration responses of the dam were obtained.”

5-       Please specify in the abstract what software you used for the modeling used?

6-       The term adaptive recursive subspace method is used a lot in the article, use its abbreviation as ARS method.

7-       Lines 24-27: The proposed adaptive recursive subspace method is used to identify the time-varying modal parameters of the arch dam, and the identification results are consistent with the evolution of damage, thus verifying the effectiveness of the proposed method.

The identification results of the ARS method are consistent with the evolution of damage verifying the  effectiveness of the proposed method.”

8-       Unfortunately, the structure of the abstract is not suitable. The key results of the research are not given and only a general conclusion is given in the last sentence. The abstract should be revised according to the mentioned points  and should be a total of about 200 words maximum. 

9-      English language correction is required.

10-  Line 42-44: Please check the grammar.

11-  Line 61: Correct “This work holds significant theoretical significance and practical value….”

12-  For all references put the space between the noun and  all of the Square brackets, flood resistance[1-3]

….. space [1-3]

13-  Lines 65 to 100: Put it in 3 or 4 paragraphs and express it more fluently.

14- Lines 69 to 74 are grammatically wrong, especially about punctuation.

15-  From lines 69 to 100, for all of the text give references. If a paragraph has no reference omit it.

16-  Lines 101 to 132: Put it in 3 or 4 paragraphs and express it more fluently.

17-  From lines 104 to 120, for all of the text give references. If a paragraph has no reference omit it.

18-  Lines 123 to 132,  They are given exactly in the abstract and are completely repeated. In this part, the main work done in the research should be presented in a text different from the abstract in such a way that the innovation of the research is brought into it. Explain the numerical models of “A concrete plastic damage model” and “numerical simulation of a concrete arch dam” in more detail. It is very important to bring the research innovation in detail in this section.

19-  Unfortunately, in this article, according to the guidelines, the parts of the article including the materials and materials, which are the same as the second part and the findings of the third part of the article, are not given. After the results, the authors have given the conclusions. Before that, the authors did not discuss and compare the results with most of the studies. Every finding that the authors have obtained from the analysis as a result of modeling should be challenged and compared with the findings of the articles in this field. Without the discussion section, the paper is similar to the text in which the authors report the findings of their analysis.

20-  Bring all the symbols used in the entire text of the article in the form of a table at the end of section 2.

21-  From the beginning until we reach line 251, in none of the previous sections of the article, including the abstract and introduction, the exact specifications of the used model are not clear, for example

"A three-dimensional dynamic finite element simulation analysis was conducted on a specific arch dam"

Instead of repeating the last paragraph of the introduction from the abstract, explain finite element modeling.

22-  In a comment, explain in detail what software you used for numerical modeling and whether these software have a valid license also in the text of article.

23-  Line 266: The abbreviation should be used in the first place where the full phrase concrete plastic damage appears, where the full phrase is used for the first time in line 22.  

24-  Line 270: CDP or CPD?

25-  Line 272: remove CPD

26-  Line 273: This is an inappropriate way of referring.

The material parameters for the dam body and foundation, along with the parameters of the concrete plasticity damage constitutive model for the dam body and the relationship curves of the CPD constituent model, are according to the study by J et al. [8].

27-  Line 344: Please give an explanation for N4SID in parentheses.

28-  Line 332, 366,  Hz is inappropriately repeated several times.

29-  Line 321,364,368  % is inappropriately repeated several times.

30-  The conclusion section is too long, which should be shortened in the revised manuscript.

Comments on the Quality of English Language

Moderate editing of English language required

Author Response

Thank you very much for your comments on our paper. Your suggestions truly helped us improve the quality of this paper. The manuscript has been revised according to your comments. All the modified content has been marked in red in the revised manuscript. Some of your questions are answered below.

Comment:

This is an interesting study. Timely identification of earthquake damage for arch dam concrete was done by analyzing the variations in structural dynamic properties measured by the dynamic response of concrete dams.

According to the iThenticate report, the similarity rate is 28% and there are many similarities in parts of the abstract and introduction and other parts of this article about the main work done in this research, please explain the difference with the article “Online structural damage state identification of concrete arch dams under dynamic loads using a recursive TVARX approach” completely in explain the introduction. Bring the innovation of this research  compare to the article “Online structural damage state identification of concrete arch dams under dynamic loads using a recursive TVARX approach” in the form of comments, in the abstract, and the introduction.

I believe rather than just reporting the results, it is important to provide a critical discussion of the findings. At the moment, the paper mostly looks like a report rather than a scientific paper. The results and discussions have few discussions based on current literature and comparisons among other papers. Therefore, a major revision is required before accepting the paper for publication. 

Other minor and major suggestions are described below:

1-      Time-varying Modal Parameters Identification of Concrete Arch Dam Using an Adaptive Recursive Subspace Method

Time-varying Modal Parameters Identification…. many of words including nouns and adjectives are listed. I suggest “Identification of Time-varying Modal Parameters for Concrete Arch Dam Using an Adaptive Recursive Subspace Method”

Answer: Thank you for your valuable suggestion. The title of the article has been revised to “Identification of Time-varying Modal Parameters for Concrete Arch Dam Using an Adaptive Recursive Subspace Method”.

2-      All the verbs that express the research done by the authors should be expressed in the past tense.

 “time-varying modal parameters”

This term is repeated a lot in the text of the article, please abbreviate it as "TMP".

Answer: Thank you for the two valuable suggestions. The verbs that express the research done by the authors have been revised and expressed in the past tense, and “time-varying modal parameters” in this article has been abbreviated as "TMP".

3-       from lines 16 to 20,  “the identification of the time-varying modal parameters of concrete arch dams”.

It has been repeated redundantly.

“ In this study, the identification of the time-varying modal parameters of arch dams under seismic loads is studied.”

I suggest “ In this study, the identification of the time-varying modal parameters of arch dams under seismic loads was a challenge that an adaptive recursive subspace method with variable forgetting factors used to solve it.” that the authors could change it.

Answer: We have adopted this valuable suggestion.

4-       line 20: To validate the effectiveness of the proposed adaptive recursive subspace method, a numerical simulation of a concrete arch dam under seismic loading is conducted, and the acceleration responses of the dam are obtained.

You cannot use numerical modeling to verify the method, the question is how to validate and verify this numerical modeling, I suggest

“ A numerical simulation of a concrete arch dam under seismic loading was conducted, and the acceleration responses of the dam were obtained.”

Answer: We appreciate this valuable suggestion and have revised it according to your suggestion.

5-       Please specify in the abstract what software you used for the modeling used?

Answer: We appreciate this valuable suggestion and the widely used finite element analysis software ABAQUS was used for the simulation modeling.

6-       The term adaptive recursive subspace method is used a lot in the article, use its abbreviation as ARS method.

Answer: We have adopted this suggestion, and “adaptive recursive subspace method” in this article has been abbreviated as "ARS method”

7-       Lines 24-27: The proposed adaptive recursive subspace method is used to identify the time-varying modal parameters of the arch dam, and the identification results are consistent with the evolution of damage, thus verifying the effectiveness of the proposed method.

“The identification results of the ARS method are consistent with the evolution of damage verifying the effectiveness of the proposed method.”

Answer: We appreciate this valuable suggestion and have revised it according to your suggestion.

8-       Unfortunately, the structure of the abstract is not suitable. The key results of the research are not given and only a general conclusion is given in the last sentence. The abstract should be revised according to the mentioned points and should be a total of about 200 words maximum. 

Answer: We have adopted this suggestion and revised the abstract.

9-      English language correction is required.

Answer: We have corrected the language throughout the entire article.

10-  Line 42-44: Please check the grammar.

Answer: We have checked Line 42-44, and ‘To date, although there have been few major failure cases for concrete dams in seismic events, several structural damage cases for gravity dams caused by earthquakes have occurred, such as those of the Koyna Dam’ has been revised as ‘Several instances of dams sustaining damage due to strong earthquakes have been recorded, including the Hsinfengkiang concrete dam [6] in China, the Koyna gravity dam [7] in India, and the Pacoima arch dam [8] in the USA.’

11-  Line 61: Correct “This work holds significant theoretical significance and practical value….”

Answer: We have corrected this sentence.

12-  For all references put the space between the noun and  all of the Square brackets, flood resistance[1-3]

….. space [1-3]

Answer: We have adopted this valuable suggestion.

13-  Lines 65 to 100: Put it in 3 or 4 paragraphs and express it more fluently.

Answer: We have adopted this valuable suggestion, and 3 paragraphs have been created. 

14- Lines 69 to 74 are grammatically wrong, especially about punctuation.

Answer: We have checked Lines 69-74, and they are revised.

15-  From lines 69 to 100, for all of the text give references. If a paragraph has no reference omit it.

Answer: We have adopted this suggestion and added references in this paragraph.

16-  Lines 101 to 132: Put it in 3 or 4 paragraphs and express it more fluently.

Answer: We have adopted this valuable suggestion, and 4 paragraphs have been created. 

17-  From lines 104 to 120, for all of the text give references. If a paragraph has no reference omit it.

Answer: We have adopted this suggestion and added references in the revised paragraph.

18-  Lines 123 to 132,  They are given exactly in the abstract and are completely repeated. In this part, the main work done in the research should be presented in a text different from the abstract in such a way that the innovation of the research is brought into it. Explain the numerical models of “A concrete plastic damage model” and “numerical simulation of a concrete arch dam” in more detail. It is very important to bring the research innovation in detail in this section.

Answer: We have adopted this suggestion and Lines 123 to 132 had been revised.

19-  Unfortunately, in this article, according to the guidelines, the parts of the article including the materials and materials, which are the same as the second part and the findings of the third part of the article, are not given. After the results, the authors have given the conclusions. Before that, the authors did not discuss and compare the results with most of the studies. Every finding that the authors have obtained from the analysis as a result of modeling should be challenged and compared with the findings of the articles in this field. Without the discussion section, the paper is similar to the text in which the authors report the findings of their analysis.

 Answer: We have adopted this suggestion and Lines 132 to 132 had been revised.

20-  Bring all the symbols used in the entire text of the article in the form of a table at the end of section 2.

 Answer: We have adopted this suggestion and Table 1 was created.

21-  From the beginning until we reach line 251, in none of the previous sections of the article, including the abstract and introduction, the exact specifications of the used model are not clear, for example

"A three-dimensional dynamic finite element simulation analysis was conducted on a specific arch dam"

Instead of repeating the last paragraph of the introduction from the abstract, explain finite element modeling.

 Answer: We have adopted this suggestion and the description of the used model has been modified and standardized.

22-  In a comment, explain in detail what software you used for numerical modeling and whether these software have a valid license also in the text of article.

 Answer: The finite element analysis software ABAQUS was used for the numerical modeling in this study, and the software ABAQUS has a valid license.

23-  Line 266: The abbreviation should be used in the first place where the full phrase concrete plastic damage appears, where the full phrase is used for the first time in line 22. 

 Answer: The concrete damaged plasticity (CDP) model was used to simulate the dam body’s constitutive relation, and the expression and abbreviation of the concrete damaged plasticity model had been revised and standardized throughout the entire text.

24-  Line 270: CDP or CPD?

 Answer: The concrete damaged plasticity (CDP) model had been revised.

25-  Line 272: remove CPD

Answer: We have removed CPD.

26-  Line 273: This is an inappropriate way of referring.

The material parameters for the dam body and foundation, along with the parameters of the concrete plasticity damage constitutive model for the dam body and the relationship curves of the CPD constituent model, are according to the study by J et al. [8].

Answer: Thank you for your valuable suggestion, and we have revised Line 273 according to your suggestion.

27-  Line 344: Please give an explanation for N4SID in parentheses.

Answer: We have adopted this suggestion, and N4SID is the abbreviation of numerical algorithms for subspace state space system identification.

28-  Line 332, 366,  Hz is inappropriately repeated several times.

Answer: Hz is the abbreviation of Hertz, and We have adopted this suggestion and revised it.

29-  Line 321,364,368  % is inappropriately repeated several times.

Answer: % represents a percentage, and this damping ratio expression can also be found in other literatures such as ‘ZHONG K H, CHANG C C. Tracking dynamic characteristics of structures using output-only recursive combined subspace identification technique [ J ]. 2020, 146(5): 04020035. Doi: 10.1061/(ASCE)EM.1943-7889.0001742’. Figure 1 is the partial text screenshot of the article entitled “Tracking dynamic characteristics of structures using output-only recursive combined subspace identification technique”, in which the  damping ratio values were also expressed with %.

Figure 1 the partial text screenshot of the article entitled “Tracking dynamic characteristics of structures using output-only recursive combined subspace identification technique”

30-  The conclusion section is too long, which should be shortened in the revised manuscript.

Answer: Thank you for this valuable suggestion, we have shortened the conclusion section.

 Dear Reviewer 1
    We are writing to express our sincere gratitude for your thorough review of our manuscript titled "Identification of Time-varying Modal Parameters for Concrete Arch Dams Using an Adaptive Recursive Subspace Method ".

Your insightful comments and suggestions have been immensely valuable in refining the quality of our work. We would like to inform you that we have carefully considered each of your recommendations and have incorporated them into the revised version of the manuscript. Your expertise has undoubtedly contributed to enhancing the clarity, coherence, and overall strength of our research.

We truly appreciate the time and effort you dedicated to providing constructive feedback, and we believe that your input has significantly improved the overall quality of our manuscript. Your commitment to advancing scholarly work is commendable, and we are grateful for your dedication to the peer review process.

Please find attached the revised manuscript for your perusal. We hope that the changes made align with your expectations, and we are open to any further suggestions you may have.

Once again, thank you for your invaluable contribution to our research. We look forward to hearing from you soon.

Sincerely yours,

Xinyi Zhu ,Jianchun Qiu ,Yanxin Xu ,Xingqiao Chen , Pengcheng Xu  , Xin Wu , Shaolong Guo , Jicheng Zhao , and Jiale Lin

Reviewer 2 Report

Comments and Suggestions for Authors

In the conducted study, the exploration centered on discerning the time-varying modal parameters of arch dams subjected to seismic loads. A method employing adaptive recursive subspace techniques featuring variable forgetting factors was advanced to ascertain the evolving modal characteristics of concrete arch dams. To appraise the efficacy of this proposed methodology, a numerical simulation simulating seismic loading on a concrete arch dam was executed, yielding acceleration responses crucial for analysis.

The paper was deemed to align well with the scope of the journal; however, in its current iteration, it required a Major Revision. The feedback provided was as follows:

1. The introduction failed to adequately address the research gap.

2. The introduction was inadequately written, lacking discussion on codal provisions or guidelines.

3. The literature review was overly extensive, with a suggestion to provide a brief discussion instead of simply listing papers. Recommended papers included: "Structures" Elsevier, Volume 31, pages 428-461, with a DOI of 10.1016/j.istruc.2021.01.102, and "Buildings" MDPI, Volume 13, Issue 5, with a DOI of 10.3390/buildings13051220.

4. The rationale behind the selection of the finite element size for the model was not explained.

5. There was no mention of whether the authors considered strain rate effects during material modeling, and if so, how they accounted for them.

6. The conclusions lacked quantitative data.

7. Various assumptions were made throughout the paper without proper justification. It was recommended to provide justifications for these assumptions and evaluate their potential impact on the results.

8. Lastly, recommendations stemming from the analysis were requested. Specifically, whether the authors recommended reviewing government regulations or directives for designing earthquake-resistant structures and whether any changes were deemed necessary.

A recommendation was made for a Double Major Revision. If the authors addressed all the mentioned issues and made the necessary amendments, there was a possibility for the article to be published in the MDPI journal.

Author Response

Thank you very much for your comments on our paper. Your suggestions truly helped us improve the quality of this paper. The manuscript has been revised according to your comments. All the modified content has been marked in red in the revised manuscript. Some of your questions are answered below.

Comment: In the conducted study, the exploration centered on discerning the time-varying modal parameters of arch dams subjected to seismic loads. A method employing adaptive recursive subspace techniques featuring variable forgetting factors was advanced to ascertain the evolving modal characteristics of concrete arch dams. To appraise the efficacy of this proposed methodology, a numerical simulation simulating seismic loading on a concrete arch dam was executed, yielding acceleration responses crucial for analysis.

The paper was deemed to align well with the scope of the journal; however, in its current iteration, it required a Major Revision. The feedback provided was as follows:

  1. The introduction failed to adequately address the research gap.

Answer: Thanks for this valuable suggestion, and we have revised it according to your suggestion.

The study of the recursive subspace method for real-time identification of the TMP of concrete dams, based on real-time input‒output data, allows for the continuous estimation of system matrices and modal parameters. To address efficiency concerns arising from the large data volume and increased Hankel matrix dimension during the identification process, a fixed forgetting factor mechanism is typically introduced to enhance the tracking ability for TMP [28].  While this approach has yielded certain research achievements, it lacks adaptability because it requires manual setting of the forgetting factor based on prior knowledge. To address the low efficiency in tracking TMP using conventional recursive subspace method, an ARS identification method based on variable forgetting factors was proposed in this study. Through the introduction of the variable forgetting factor, an unbiased updating form for Hankel matrix was derived. The variable forgetting factors were adaptive updated by utilizing the spatial Euclidean distance of identified modal frequencies at adjacent time instants. The Hankel matrices with variable forgetting factors, as well as the generalized observable matrices and system matrices with variable forgetting factors, were established and derived to achieve the adaptive recursion of subspace method with variable forgetting factors. Through the eigenvalue decomposition of the system state space matrix, the real-time modal parameters of concrete arch dam can be obtained.

  1. The introduction was inadequately written, lacking discussion on codal provisions or guidelines.

Answer: We have revised the Introduction section and the discussion on the research background and guidelines had been supplemented. 

  1. The literature review was overly extensive, with a suggestion to provide a brief discussion instead of simply listing papers. Recommended papers included: "Structures" Elsevier, Volume 31, pages 428-461, with a DOI of 10.1016/j.istruc.2021.01.102, and "Buildings" MDPI, Volume 13, Issue 5, with a DOI of 10.3390/buildings13051220.

Answer: We have referenced the papers you recommended, made modifications to the introduction section, and provided brief discussions on certain references.

  1. The rationale behind the selection of the finite element size for the model was not explained.

Answer: Thanks for your suggestion, we have added the explanation of the finite element size.Upon calculation, it has been determined that extending the model 2.5 times the dam height in the upstream, downstream, left, and right directions, as well as 2 times the dam height in the foundation direction, meets the precision requirements. If the model extends too far in all directions, it will result in an increased number of elements and a corresponding rise in computational load. Therefore, with the prerequisite of ensuring computational accuracy, the finite element model's extension length is chosen based on the specified distances.

  1. There was no mention of whether the authors considered strain rate effects during material modeling, and if so, how they accounted for them.

Answer: We have supplemented the strain rate effects during material modeling, and Figures 2 and 3 showed the relationship curves of the CDP constituent model.

(a)                        (b)

Figure 2 Relationship curve among compressive stress, inelastic compressive strainand compressive damage factor

(a)                        (b)

Figure 3 Relationship curve among tensile stress, cracking strain and tensile damage factor

  1. The conclusions lacked quantitative data.

Answer: The relevant explanations regarding quantitative data have been added in the conclusion.

  1. Various assumptions were made throughout the paper without proper justification. It was recommended to provide justifications for these assumptions and evaluate their potential impact on the results.

Answer: Three assumptions were made in the paper. The first assumption is the initial moment  and the current moment ,which is a reasonable assumption and had no impact on the identification result. The second assumption is a uniform failure pattern in all directions for the elements by the adopted CDP model, which is a commonly used assumption in numerical simulation and had no impact on the identification result. The third assumption is the system order in the subspace method assuming to range from 2 to 100, which is also a commonly used assumption in subspace identification method by many references, which also had no impact on the identification results.

  1. Lastly, recommendations stemming from the analysis were requested. Specifically, whether the authors recommended reviewing government regulations or directives for designing earthquake-resistant structures and whether any changes were deemed necessary.

 Answer: The research findings presented in this paper offer valuable insights and essential references for tracking the time-varying modal parameters of arch dams exposed to strong seismic effects. There are existing specifications related to seismic monitoring of concrete dams. This study, based on seismic monitoring data of arch dams, performs the identification of time-varying modal parameters. This proves to be a pivotal reference for processing seismic monitoring data of arch dams and promptly identifying damage. It carries significant importance in ensuring the long-term, efficient, and safe operation of concrete dam projects, thereby enhancing their capabilities in flood control, power generation, water supply, and other critical aspects. Furthermore, it contributes to the advancement of national economic and social development.

The research findings of this paper also hold certain reference value for revising content related to real-time structural modal parameter identification and damage diagnosis in the technical specifications for seismic monitoring of concrete arch dams. For example, according to the numerical simulation results, reinforcing seismic monitoring and damage diagnosis at the upper arch crown and dam toe positions could be considered, and the proposed method offer valuable insights and essential references for tracking the time-varying modal parameters of arch dams exposed to strong seismic effects.

  1. A recommendation was made for a Double Major Revision. If the authors addressed all the mentioned issues and made the necessary amendments, there was a possibility for the article to be published in the MDPI journal.

 Answer: Thank you very much for all your comments on our paper! We have revised our paper according to your comments.

Dear reviewer 2

We are writing to express our sincere gratitude for your thorough review of our manuscript titled "Identification of Time-varying Modal Parameters for Concrete Arch Dams Using an Adaptive Recursive Subspace Method ".

Your insightful comments and suggestions have been immensely valuable in refining the quality of our work. We would like to inform you that we have carefully considered each of your recommendations and have incorporated them into the revised version of the manuscript. Your expertise has undoubtedly contributed to enhancing the clarity, coherence, and overall strength of our research.

We truly appreciate the time and effort you dedicated to providing constructive feedback, and we believe that your input has significantly improved the overall quality of our manuscript. Your commitment to advancing scholarly work is commendable, and we are grateful for your dedication to the peer review process.

Please find attached the revised manuscript for your perusal. We hope that the changes made align with your expectations, and we are open to any further suggestions you may have.

Once again, thank you for your invaluable contribution to our research. We look forward to hearing from you soon.

Sincerely yours,

Xinyi Zhu ,Jianchun Qiu ,Yanxin Xu ,Xingqiao Chen , Pengcheng Xu  , Xin Wu , Shaolong Guo , Jicheng Zhao , and Jiale Lin

Reviewer 3 Report

Comments and Suggestions for Authors

The paper proposes a method to identify the time-varying modal parameters. The idea seems good but it is not well explained. This is a seriuos shortcomings and so the paper in this actual state cannot be accepted. Some maior issues are listed below.

1) Paragraph 2: What does the authors mean using the expression "forgetting factors"? It is not very clear. Please try to better explain and use a more intuitive word.

2) Equation 6, state sequence Z, is not clear. Please try to better explain.

3) All the paragraph 2 is not clear. It must be improved.

4) Some observations on the numerical simulation: The model should be much more described. for example nothing is said about the boundaries, the material and plasticity should be better explained (even if reported in reference 8), The values for coefficientsa0 and a1 in the rayleigh relationship have not been discussed, a modal analysis for the model is advisable to have an idea about the expected seismic response as well as the spectrum of the seismic excitation, this latter need to be better specified (artificial or natural).

5) Line 294: "They have been identified in several finite element simulations and shaking table model tests", please provide some references.

6) Figure 7:  The Power Spectral Density (PSD) could be improved. What measuring point the psd is referred at? All PSDs could be represented in order to reduce the uncertainties on the identification.

7) In the numerical simulations is not clear if the forgetting factors change during the simulation or remain the same. How are they selected initially?

Comments on the Quality of English Language

Moderate editing of English language required

Author Response

Thank you very much for your comments on our paper. Your suggestions truly helped us improve the quality of this paper. The manuscript has been revised according to your comments. All the modified content has been marked in red in the revised manuscript. Some of your questions are answered below.

Comment: The paper proposes a method to identify the time-varying modal parameters. The idea seems good but it is not well explained. This is a seriuos shortcomings and so the paper in this actual state cannot be accepted. Some maior issues are listed below.

1) Paragraph 2: What does the authors mean using the expression "forgetting factors"? It is not very clear. Please try to better explain and use a more intuitive word.

Answer: Thanks for your valuable suggestion, we have added the explanation about "forgetting factors".

 In the recursive identification process of TMP using the RS method, a duration of input-output response data is necessary. As the recursive identification progresses, new input-output data obtained at each moment is consistently integrated, while old in-put-output data is systematically removed and disregarded. Consequently, the structural modal parameters are continuously refined at each moment. It is evident that data from the current moment and those closer to it are more reflective of the current state compared to data from further back. Therefore, in the recursive subspace method, a forgetting factor is often introduced to express the varying impact of data from different time points on the current moment, distinguishing between newer and older data. However, a fixed forgetting factor is commonly employed for slow-varying systems [33]. Concerning the selection of the forgetting factor, a larger value can expedite convergence in slow-varying systems; nevertheless, in the presence of a substantial change in the system subspace, the tracking performance of these methods may experience a notable decline. On the contrary, a relatively smaller forgetting factor can enhance the tracking capability of fast-varying parameters. Consequently, a variable forgetting factor recursive subspace algorithm has been developed to proficiently track systems with both slow and fast-varying parameters.

Forgetting factor is a commonly used word in recursive subspace method, and some references such as the following references also used this word.  

Nguyen TD, Yamada I. Necessary and sufficient conditions for convergence of the DDT systems of the Normalized PAST algorithms [J]. Signal Processing. 2014(94): 288-299.

Zhiyu Ni, Shunan Wu, Chenchen Wu. Time-varying frequency parameter identification of space solar power satellites based on an improved recursive subspace algorithm and optimal sensor placement [J]. Aerospace Science and Technology. 2022(128):107754.

2) Equation 6, state sequence Z, is not clear. Please try to better explain.

Answer: Thanks for your valuable suggestion. is a general state vector sequence with forgetting factors, and  designated the state vector composed of displacement and velocity at time .

3) All the paragraph 2 is not clear. It must be improved.

Answer: We have revised and improved paragraph 2.
4) Some observations on the numerical simulation: The model should be much more described. for example nothing is said about the boundaries, the material and plasticity should be better explained (even if reported in reference 8), The values for coefficientsa0 and a1 in the rayleigh relationship have not been discussed, a modal analysis for the model is advisable to have an idea about the expected seismic response as well as the spectrum of the seismic excitation, this latter need to be better specified (artificial or natural).

Answer: Thanks for these valuable suggestions. The introduction of the numerical model has been supplemented and revised. Rayleigh damping  is used for the material damping form of the dam body, in which and , and and designate the mass matrix and stiffness matrix, respectively. The damping coefficients  and  are obtained from the following two equations:

Where  is the first-order natural circular frequencies of the arch dam,  is the fifth-order natural circular frequencies of the arch dam, and  and  are the damping ratios corresponding to  and , respectively.

5) Line 294: "They have been identified in several finite element simulations and shaking table model tests", please provide some references.

Answer: They ( the two damaged locations shown in Figure 5) have also appeared in multiple finite element simulations and shaking table model tests of arch dams, and some references had been provided.

6) Figure 7:  The Power Spectral Density (PSD) could be improved. What measuring point the psd is referred at? All PSDs could be represented in order to reduce the uncertainties on the identification.

Answer: The power spectral density in Figure 7 referred at the average of all measuring points. The amplitude differences of the power spectral density at each measurement point are relatively large, but not prominently displayed in a single graph. Considering that the power spectral density is not the focus of this study, only the average power spectral density of each measurement point during this time period is presented in the figure.

7) In the numerical simulations is not clear if the forgetting factors change during the simulation or remain the same. How are they selected initially?

Answer: The numerical simulation model in this study replicates the occurrence and progression of damage in a 305-meter arch dam subjected to a seismic load. This model validates the proposed adaptive recursive subspace method presented in this paper. Notably, forget factors were not applied in the numerical simulation; however, they play a crucial role in the proposed adaptive recursive subspace algorithm.

In the recursive identification process of TMP using the RS method, a duration of input-output response data is necessary. As the recursive identification progresses, new input-output data obtained at each moment is consistently integrated, while old in-put-output data is systematically removed and disregarded. Consequently, the structural modal parameters are continuously refined at each moment. It is evident that data from the current moment and those closer to it are more reflective of the current state compared to data from further back. Therefore, in the recursive subspace method, a forgetting factor is often introduced to express the varying impact of data from different time points on the current moment, distinguishing between newer and older data. However, a fixed forgetting factor is commonly employed for slow-varying systems [33]. Concerning the selection of the forgetting factor, a larger value can expedite convergence in slow-varying systems; nevertheless, in the presence of a substantial change in the system subspace, the tracking performance of these methods may experience a notable decline. On the contrary, a relatively smaller forgetting factor can enhance the tracking capability of fast-varying parameters. To address the low efficiency in tracking TMP using conventional RS method, an ARS identification method based on variable forgetting factors is was proposed in this study.

Dear reviewer 3

We are writing to express our sincere gratitude for your thorough review of our manuscript titled "Identification of Time-varying Modal Parameters for Concrete Arch Dams Using an Adaptive Recursive Subspace Method ".

Your insightful comments and suggestions have been immensely valuable in refining the quality of our work. We would like to inform you that we have carefully considered each of your recommendations and have incorporated them into the revised version of the manuscript. Your expertise has undoubtedly contributed to enhancing the clarity, coherence, and overall strength of our research.

We truly appreciate the time and effort you dedicated to providing constructive feedback, and we believe that your input has significantly improved the overall quality of our manuscript. Your commitment to advancing scholarly work is commendable, and we are grateful for your dedication to the peer review process.

Please find attached the revised manuscript for your perusal. We hope that the changes made align with your expectations, and we are open to any further suggestions you may have.

Once again, thank you for your invaluable contribution to our research. We look forward to hearing from you soon.

Sincerely yours,

Xinyi Zhu ,Jianchun Qiu ,Yanxin Xu ,Xingqiao Chen , Pengcheng Xu  , Xin Wu , Shaolong Guo , Jicheng Zhao , and Jiale Lin

Round 2

Reviewer 1 Report

Comments and Suggestions for Authors

All comments have been satisfied. I appreciate the authors' efforts.

Author Response

Thanks for your  recognition and acceptance of the revised manuscript. Your insightful feedback and constructive comments played a pivotal role in enhancing the quality and clarity of the manuscript. Your expertise in the field has undoubtedly enriched the content, and I am truly appreciative of the time and effort you invested in reviewing my work.  I am genuinely honored to have received your approval. 

Best regards,

Xinyi Zhu, Jianchun Qiu, Yanxin Xu, et al

Reviewer 2 Report

Comments and Suggestions for Authors

The reviewer acknowledges the efforts made by the authors in revising the manuscript. However, the manuscript has not yet reached a stage where a decision can be made. Consequently, this reviewer remains unsatisfied with the authors' responses and their incorporation of suggestions. It is imperative that the authors take the comments seriously and integrate them into the manuscript. Specifically:

1. The abstract currently lacks appeal to readers. It should be more concise and include some quantitative results highlighting the study findings.

2. The introduction still fails to adequately address the research gap. It should incorporate design codal provisions and requirements.

3. While the literature review has been improved, it is not yet at the level required for acceptance in a reputable scientific journal. Therefore, it needs to be carefully reviewed and enhanced. Suggested papers included: DOI: 10.1016/j.istruc.2021.01.102.

4. The study's incorporation of strain rate effects needs clarification. If not addressed, it should be specified in the manuscript.

This reviewer suggests minor revisions to address these concerns.

Author Response

Thank you very much for your comments on our paper. Your suggestions truly helped us improve the quality of this paper. The manuscript has been revised according to your comments. All the modified content has been marked in red in the revised manuscript. Some of your questions are answered below.

Comment: The reviewer acknowledges the efforts made by the authors in revising the manuscript. However, the manuscript has not yet reached a stage where a decision can be made. Consequently, this reviewer remains unsatisfied with the authors' responses and their incorporation of suggestions. It is imperative that the authors take the comments seriously and integrate them into the manuscript. Specifically:

  1. The abstract currently lacks appeal to readers. It should be more concise and include some quantitative results highlighting the study findings.

Answer: Thanks for this valuable suggestion, and we have revised the abstract according to your suggestion.

  1. The introduction still fails to adequately address the research gap. It should incorporate design codal provisions and requirements.

Answer: We have revised the Introduction section and the discussion on the research background and guidelines had been supplemented.

To address the low efficiency of the conventional RS method in tracking TMP of concrete arch dams, an ARS method based on variable forgetting factors was proposed in this study. In the ARS method, the variable forgetting factors were adaptive updated by assessing the changing rate of the spatial Euclidean distance of adjacent modal frequency identification values. Through the introduction of the variable forgetting factor, an unbiased updating form for the Hankel matrix was derived. The Hankel matrices with variable forgetting factors, as well as the generalized observable matrices and system matrices with variable forgetting factors, were established and derived to achieve the adaptive recursion of subspace method with variable forgetting factors. The recursive updating of the generalized observability matrix was realized through the Orthogonal Subspace Tracking algorithm, thereby enabling the recursive identification of the system's state-space matrix. From the eigenvalue decomposition of the system's state-space matrix, the identification of time-varying modal parameters was achieved.

We also explored some design or technical codes related to the seismic monitoring of arch dams, and these explanation can be seen in the revised manuscript. On the other hand, the research outcomes of this paper offer a computational approach and methodology for determining the time-varying modal parameters of arch dams, providing scientific basis for related technical specifications.

The discussion about the design or technical codes was given as follows:

Structural damage identification of concrete arch dams based on dynamic characteristics possesses the advantages of being non-destructive, multi-level, and multi-scale. To achieve the goal of damage identification of arch dams using dynamic characteristics, seismic monitoring should be conducted for arch dams. In recent years, the seismic monitoring for arch dams have been proposed by some design cods. For example, the code entitled ‘Technical Specification of Strong Motion Monitoring for Seismic Safety of Hydraulic Structures’ (Guideline number SL 486-2011) in China provided clear provisions for the arrangement and operational management of strong seismic monitoring instruments for arch dams in seismic high-intensity areas. Meanwhile, the code entitled 'Technical Specifications for Safety Monitoring of Concrete Dams (Guideline number SL601-2013)' in China mandated that dynamic response monitoring instruments must be installed in Grade 1 dams designed for a seismic intensity of VII or higher, and in Grade 2 dams designed for a seismic intensity of VIII or higher. Following the requirements of these technical specifications, there has been a growing trend among concrete arch dams in high-seismic areas to implement the setup of seismic observation instruments.

  1. While the literature review has been improved, it is not yet at the level required for acceptance in a reputable scientific journal. Therefore, it needs to be carefully reviewed and enhanced. Suggested papers included: DOI: 10.1016/j.istruc.2021.01.102.

Answer: Thanks for your valuable suggestion. We have conducted a thorough and detailed review and revision of the literature review, including enhancements such as additional references and supplementary discussions of some cited works.

  1. The study's incorporation of strain rate effects needs clarification. If not addressed, it should be specified in the manuscript.

Answer: The CDP model was constructed by integrating plastic mechanics with damage mechanics and has been incorporated into the computational analysis framework of the software ABAQUS. This model utilizes the yield criteria from plastic mechanics to determine whether concrete has entered a state of damage. If the material is in a damaged state, the flow rule is applied to calculate the post-damage inelastic strain. Subsequently, the stiffness degradation of the material after damage is calculated based on the damage evolution curve. By establishing a relationship between inelastic strain and the damage factor, the model facilitates the simulation of various degrees of structural damage.

Dear reviewer 2

We are writing to express our sincere gratitude for your second review of our manuscript titled "Identification of Time-varying Modal Parameters for Concrete Arch Dams Using an Adaptive Recursive Subspace Method ".

Your insightful comments and suggestions have been immensely valuable in refining the quality of our work. We would like to inform you that we have carefully considered each of your recommendations and have incorporated them into the revised version of the manuscript. Your expertise has undoubtedly contributed to enhancing the clarity, coherence, and overall strength of our research.

We truly appreciate the time and effort you dedicated to providing constructive feedback, and we believe that your input has significantly improved the overall quality of our manuscript. Your commitment to advancing scholarly work is commendable, and we are grateful for your dedication to the peer review process.

Please find attached the revised manuscript for your perusal. We hope that the changes made align with your expectations, and we are open to any further suggestions you may have.

Once again, thank you for your invaluable contribution to our research. We look forward to hearing from you soon.

Sincerely yours,

Xinyi Zhu ,Jianchun Qiu ,Yanxin Xu ,Xingqiao Chen , Pengcheng Xu  , Xin Wu , Shaolong Guo , Jicheng Zhao , and Jiale Lin
